# Prediction of storm transfers and annual loads with data-based mechanistic models using high-frequency data

Mary C.Ockenden[1], Wlodek Tych[1], Keith J.Beven[1], Adrian L.Collins[2], Robert Evans[3], Peter D.Falloon[4], Kirsty J.Forber[1], Kevin M.Hiscock[5], Michael J.Hollaway[1], Ron Kahana[4], Christopher J.A.Macleod[6], Martha L.Villamizar[7], Catherine Wearing[1], Paul J.A.Withers[8], Jian G.Zhou[9], Sean Burke[10], Richard J.Cooper[5], Jim E.Freer[11], Philip M.Haygarth[1]

[1]Lancaster Environment Centre, Lancaster University, Bailrigg, Lancaster LA1 4YQ, England, UK
[2]Rothamsted Research North Wyke, Okehampton, Devon EX20 2SB, England, UK
[3]Global Sustainability Institute, Anglia Ruskin University, Cambridge CB1 1PT, England, UK
[4]Met Office Hadley Centre, Exeter, Devon EX1 3PB, England, UK
[5]School of Environmental Sciences, University of East Anglia, Norwich NR4 7TJ, England, UK
[6]James Hutton Institute, Aberdeen AB15 8QH, Scotland, UK
[7]School of Engineering, Liverpool University, Liverpool L69 3GQ, England, UK
[8]Bangor University, Bangor, Gwynedd LL57 2UW, Wales, UK
[9]School of Computing, Mathematics & Digital Technology, Manchester Metropolitan University, Manchester M1 5GD, UK
[10]British Geological Survey, Keyworth, Nottingham NG12 5GG, England, UK
[11]University of Bristol, Bristol BS8 1SS, UK

*Correspondence to*: Philip M. Haygarth  (p.haygarth@lancaster.ac.uk)

**Abstract.** Excess nutrients in surface waters, such as phosphorus (P) from agriculture, result in poor water quality, with adverse effects on ecological health and costs for remediation. However, understanding and prediction of P transfers in catchments have been limited by inadequate data and over-parameterised models with high uncertainty. We show that, with high temporal resolution data, we are able to identify simple dynamic models that capture the P load dynamics in three contrasting agricultural catchments in the UK. For a flashy catchment, a linear, second-order (two pathways) model for discharge gave high simulation efficiencies for short-term storm sequences and was useful in highlighting uncertainties in out-of-bank flows. A model with non-linear rainfall input was appropriate for predicting seasonal or annual cumulative P loads where antecedent conditions affected the catchment response. For second-order models, the time constant for the fast pathway varied between 2 and 15 hours for all three catchments and for both discharge and P, confirming that high temporal resolution data are necessary to capture the dynamic responses in small catchments ($10\text{-}50 \ km^2$). The models led to a better understanding of the dominant nutrient transfer modes, which will be helpful in determining phosphorus transfers following changes in precipitation patterns in the future.

## 1 Introduction

The quality of both surface waters and groundwater is under increasing pressure from numerous sources, including intensive agricultural practices, water abstraction, climate change and changes in food production and housing provision to cope with population growth (Carpenter and Bennett, 2011). Sediment and nutrient concentrations and loads are of concern to water utility companies and to environmental regulators who are striving to meet stringent water quality standards. However, accurate estimation of loads requires accurate, high temporal resolution measurements of both discharge and nutrient concentrations (Johnes, 2007) and should include quantification of observational uncertainties (McMillan et al., 2012). Sediment and nitrogen are frequently and relatively easily measured in-situ. In contrast, phosphorus (P) concentrations for water quality assessments are typically measured by manual or automatic sampling followed by laboratory analysis, often at monthly resolution, which do not capture the dynamic nature of P concentrations, and result in biased estimates of P load (Cassidy and Jordan, 2011). Phosphorus concentration in rivers and streams is controlled by many factors, including rainfall, runoff, point sources, diffuse inputs and in-stream P retention and processing. Some of these factors, particularly for small catchments, change at timescales of minutes to hours, and thus the dynamics of P concentration and load need to be studied at similar time scales. In this study, hourly time series of rainfall, runoff and P concentrations are used to help understand hydrological transport pathways of P for three contrasting agricultural catchments across the UK.

There is a wide range of complexity in hydrological and water quality models, applicable at a range of scales and for different purposes. In most models there is a balance between practical simplifications and model complexity, which depends on catchment size, knowledge (or lack of) of the hydrological processes, data availability and computing power. Some of the less complex models for diffuse pollution include export coefficient models (Johnes, 1996) and the Phosphorus Indicators Tool (PIT) (Heathwaite et al., 2003; Liu et al., 2005). The most complex water quality models are idealised, process-based representations of our best understanding of reality, with a highly complex, fixed structure and many parameters, for which there is often little or no site specific data (Dean et al., 2009). These models often include a component for sediment-bound P, where the sediment transfer is based on a form of the Universal Soil Loss Equation (USLE), which is a semi-empirical model known to perform poorly (Evans and Boardman, 2016). Results generated by such process-based models are often highly uncertain, due to the uncertainty in both the model parameters and the model structure (Parker et al., 2013; Jackson-Blake et al., 2015). A review of pollutant loss studies using one process-based model, the Soil Water Assessment Tool (SWAT), revealed that most applications used a monthly time step for calibration, with few applications using a daily time step and none using a sub-daily time step. Model fit for total P (TP) concentration, measured by Nash Sutcliffe Efficiency, often exceeded 0.5 but could be as low as -0.08 for daily calibration. Depending on the calibration criteria, there may be many different parameter sets that fit the calibration data equally well, but because of a lack of data on internal variables, the models do not necessarily fit for the right reasons. Moriasi et al. (2007) advised using several different criteria for assessment of model fit, including a graphical assessment as well as quantitative metrics.

However, complex process-based models still often fail to meet the acceptance criteria (Jackson-Blake et al., 2015), even when these are relaxed to account for additional uncertainties in the measured input data (Harmel et al., 2006) such as those due to sampling method, sample storage or fractionation (Jarvie et al., 2002). Less complex process-based models, with fewer parameters, have also been developed for phosphorus transfer and have been applied with reasonable success to specific catchments, (e.g. Dupas et al., 2016; Hahn et al., 2013). Both these studies related to small catchments ($< 10 \text{ km}^2$); it was recognised that the models would only be applicable to locations where the assumptions of the model were satisfied, which is consistent with the concept of 'uniqueness of place' (Beven, 2000).

Hydrological models are subject to uncertainties in structure, parameters and measurement data (both input and output observations) (Krueger et al., 2010), and understanding the errors in measurement data is a pre-requisite to better understanding of the other uncertainties in modelling (McMillan et al., 2012). Young et al. (1996) recommended constructing models that capture the dominant modes of a system, with as few tuneable parameters as possible. Transfer function models, whose structure and parameters are determined by the information in the data, are considered to be among the most parsimonious for rainfall-flow relationships (McGuire and McDonnell, 2006; Young, 2003). Data-Based Mechanistic (DBM) modelling, which uses time-series data and fits a range of transfer functions, allows the structure of the model to be determined by the information in the monitoring data. There will still be structural errors in a DBM model, as it tries to represent a continuum of flow pathways with just the dominant modes, but this simplification will be determined by the information in the data rather than being pre-selected. This assists in getting the right answers for the right reasons (Kirchner, 2006). In contrast, there is a danger in process-based models that one might fit quite different model structures or parameter sets to the available data, i.e. the equifinality problem (Beven, 2006;Beven and Freer, 2001). An optimal DBM model and associated parameters are identified using statistical measures, but a model is only accepted if it has a plausible physical explanation (Young, 1998, 2003; Young and Beven, 1994; Young et al., 2004). With the increasing availability of high temporal resolution datasets for additional variables alongside stream discharge (Bieroza and Heathwaite, 2015; Bowes et al., 2015; Halliday et al., 2015; Outram et al., 2014), this technique has been used effectively for relating rainfall to hydrogen ion concentration in rivers (Jones and Chappell, 2014), and rainfall to dissolved organic carbon (Jones et al., 2014).

The aim of this study was to investigate, for the first time, whether simple dynamic models of P load could be identified to help understand the hydrological P processes within three contrasting agricultural catchments in the UK that represent a range of climate, topography, soil and farming types. Specifically, the objectives were:

- To identify rainfall-runoff models for each catchment, from hourly time series data collected over three years
- To develop models of P load exported from each catchment, using hourly time series data of P concentrations measured with in-situ bankside analysers

- To improve understanding of the dominant modes of catchment response through comparison of rainfall-runoff and rainfall-TP load models for each catchment.

If successful, this would be the first time that DBM modelling has been applied to high-resolution phosphorus data in catchment science.

5 ## 2 Methodology

### 2.1 Study sites

Three rural catchments with different temperate climate, topography and farm types were monitored at high-temporal resolution as part of the UK Demonstration Test Catchment (DTC) programme (Lloyd et al., 2016a; Lloyd et al., 2016b; Outram et al., 2014; McGonigle et al., 2014). These were: Newby Beck at Newby, Eden catchment, Cumbria (54.59° N, 10 2.62° W; 12.5 km²); Blackwater at Park Farm, Wensum catchment, Norfolk (52.78° N, 1.15° E; 19.7 km²); Wylye at Brixton Deverill, Avon catchment, Hampshire (51.16° N, 2.19° W; 50.2 km²) (Fig. 1). Further details of these catchments are available in SI Table S1.

### 2.2 Data collection

Rainfall was measured at 15 minute resolution at three sites in each of the Newby Beck and Blackwater catchments (Outram 15 et al., 2014; Perks et al., 2015) and summed to give hourly totals. The hourly totals from the different rain gauges were combined by areal weighting to give an hourly time series for the catchment. For the Wylye catchment, only daily rainfall was available for sites within the catchment, so raw tipping bucket data were obtained for several sites outside the catchment and analysed to produce an hourly time series which was considered most representative of the rainfall in the catchment. Further details of the rainfall analysis for the Wylye catchment are given in SI Section S1.

River water level was measured at 15 minute resolution in the three catchments, with rating curves developed for discharge estimation (Outram et al., 2014; Perks et al., 2015; Lloyd et al., 2016b). Total phosphorus (TP) concentration was determined in-situ at 30 minute intervals with a Hach Lange combined Sigmatax sampling module and Phosphax analyser using acid digestion and colorimetry (Jordan et al., 2007; Jordan et al., 2013; Perks et al., 2015). Total P loads for each hour 25 were determined by multiplying discharge (averaged to 30 minute resolution) by TP concentration for each 30 minutes and summing to give hourly totals:

$$TPload(t) = k \sum_j Q_j C_j \qquad (1)$$

where *TPload(t)* is the load (kg) exported during the hourly timestep which ends at time *t*, $Q_j$ are the discharge observations (m³s⁻¹) within the hourly timestep, $C_j$ are the corresponding TP concentration observations (mg L⁻¹) within the hourly 30 timestep, and k is a constant (= 3.6) for conversion of units to give load in kg. Visual inspection of the data indicated that aggregation of the data from 15 or 30 minute resolution to hourly did not result in a significant loss of information. This

would not be the case for very small catchments or those where the dynamics being investigated were very fast. Calculation of the load according to Eq. 1 assumes that the TP is well-mixed in the water and that the Hach Lange sampler is taking a representative sample. It also assumes that the rating curve is appropriate over the full range of stage recordings made, and that the relationship between stage and discharge is stationary. Total phosphorus load, rather than concentration, was modelled because water utility companies are concerned about the total load which may have to be removed and because both water flow and load are fluxes, so comparisons between the two are easier to interpret directly than for concentration, which is a state rather than a flux (Jones et al., 2014).

## 2.3 Transfer function model identification

Transfer function models relating the input (here, a time series of rainfall, *R*) to the output (here, a time series of either discharge, *Q*, or phosphorus load, *TPload*) were identified using continuous-time models (Young and Garnier, 2006) where possible, or in cases where data were missing or identification was difficult, with discrete time models (Young, 2003), the estimation of which handles missing data more robustly. Continuous time models are more numerically robust and have a direct interpretation as systems of differential equations (Young, 2011). Models were identified using the *RIVCBJ identification* algorithm (Refined Instrumental Variable Continuous-time Box-Jenkins identification, for continuous-time models), or *RIVBJ identification* (Refined Instrumental Variable Box-Jenkins identification for discrete-time models) that are part of the CAPTAIN toolbox (Taylor et al., 2007) for MATLAB®.

The identification algorithm always includes a noise model, by default this assumes normally distributed, uncorrelated errors, but auto-regressive moving average (ARMA) structure can be specified. The Gaussian noise model still results in asymptotically unbiased parameter estimates, but not necessarily the most statistically efficient (close to minimum variance) (Taylor et al., 2007). In this study, models up to third order were considered initially, but higher order models showed no advantage, so only models up to second order were considered in subsequent evaluations. Full models (input-output (I-O) plus ARMA structured residual noise) were assessed initially and overall they did not produce better results in all cases; therefore, in order to keep a consistent approach for all catchments, structured noise models were not specified in later model identification, In addition, transfer function models with a structured noise component generally do not improve longer term predictions of processes which are I-O dominated. The residuals structure was not strong/enough for a structured noise model to improve the model fit consistently. If there was a strong structure in the residuals, it would suggest that something was being missed in the DBM system representation. The time delay constants were estimated from the data at the same time as the model structures.

Continuous-time and discrete-time model structures are described below (from Ockenden et al., 2017). The parameter estimates in both continuous-time models and discrete-time models are formulaically related (SI Table S3).

A second-order discrete linear transfer function, denoted by [2, 2, δ] takes the form:

$$y(t) = \frac{b_1 + b_2 z^{-1}}{1 + a_1 z^{-1} + a_2 z^{-2}} u(t - \delta) + \xi_t \quad\quad\quad (2)$$

where $y(t)$ is model output at time $t$, $u(t)$ is model input, $z^{-1}$ is the backwards step operator i.e. $z^{-1}y(t) = y(t-1)$. $b_1, b_2, a_1, a_2$ are parameters determined during model identification, δ is the number of time steps of pure time delay and $\xi_t$ represents the uncertainty arising from a combination of measurement noise, other unmeasured inputs and modelling error. For a physical interpretation, second order models were only accepted it they could be decomposed by partial fraction expansion into two first order transfer functions with structure [1, 1, δ] representing fast and slow pathways, with characteristic time constants and steady state gains, i.e.

$$y(t) = \frac{b_f}{1 - a_f z^{-1}} u(t - \delta) + \frac{b_s}{1 - a_s z^{-1}} u(t - \delta) + \xi_t \quad\quad\quad (3)$$

where $b_f$ and $b_s$ are gains on the fast and slow pathways, respectively, and $a_f$ and $a_s$ are parameters characterising the time constants of the fast and slow pathways respectively. $a_f$ and $a_s$ are roots of the denominator polynomial in the second order transfer functions above (Eq. 2). This can be interpreted as two parallel linear storages.

In continuous-time, a transfer function model with time delay $\tau$ has the form:

$$Y(s) = \frac{B(s)}{A(s)} e^{-s\tau} U(s) + E(s) \quad\quad\quad (4)$$

where $Y(s)$, $U(s)$ and $E(s)$ represent the Laplace transforms of the output, input and noise, respectively. $A(s)$ and $B(s)$ represent the denominator and numerator polynomials in the derivative operator $s = \frac{d}{dt}$ that define the relationship between the input and the output, and $\tau$ represents the time delay. Second order models were only accepted if they could be decomposed by partial fraction expansion into two parallel, first-order transfer functions, i.e.

$$TPload = \frac{b_f}{s + a_f} e^{-s\tau} R + \frac{b_s}{s + a_s} e^{-s\tau} R + E \quad\quad\quad (5)$$

This can be interpreted as two parallel stores, which are depleted at different rates, determined by the time constants (direct reciprocals of $a_f$ and $a_s$) of the fast and slow components of the response, respectively. $b_f$ and $b_s$ are parameters that determine the gain of the fast and slow components, respectively. The terms 'fast' and 'slow' are used here as qualitative terms, since they are not necessarily related to specific process mechanisms; for a second order model (two stores), one store simply depletes at a slower rate than the other. Time constants are catchment specific; for example, for a first order rainfall-

runoff model which identifies just the dominant mode (one pathway), the time constant can vary from less than an hour (e.g. for a small, flashy catchment in Malaysian Borneo (Chappell et al., 2006)) to more than three months (e.g. for a chalk stream in Berkshire, UK (Ockenden and Chappell, 2011)).

This method of model identification requires high-temporal-resolution data that capture the dynamic response to the driving input; therefore, it cannot work if input data (in this case, rainfall) are missing, and does not perform well if too much output data (in this case, discharge or TPload) are missing or not showing a response.  For the Newby Beck catchment, linear models were identified for short storm sequences up to one month, and were considered applicable to periods of similar conditions.  These short-term models had a simple linear structure and very few parameters (five for a second order model). As this paper is evaluating a methodology, successful modelling over different time scales can be used as validation of the approach. Models were not identified for short periods for Blackwater and Wylye, as the presence of a much slower pathway (with a time constant of the same order as the length of the identification period) did not allow model parameter estimates to be sufficiently constrained over such short periods.

For longer time series, when seasonal change and antecedent wetness are expected to have an impact on the response, linear models were improved by inclusion of the rainfall-runoff non-linearity (Beven, 2012) based on the storage state of the catchment, for which the discharge is used as a proxy, i.e.

$$Re(t) = R(t)(Q(t-1))^\beta \tag{6}$$

where $Re(t)$ is the effective rainfall at time $t$, $R$ is the observed rainfall, $Q$ is the observed discharge and $\beta$ is a constant exponent that is optimized from the observed data at the same time as model identification.  Using a simple nonlinear function (with a single and optimised parameter) of recent discharge measurement as catchment wetness surrogate has been tested on catchments of different size and nature, (e.g. Beven, 2012; Chappell et al., 1999; McIntyre and Marshall, 2010; Young, 2003; Young and Beven, 1994).  A recent high flow will be highly correlated with high 'overall' catchment wetness, and using the flow at time $t$-1, as in Eq. 6, still allows estimation of $Re$ and $Q$ at time t.  The resulting effective inputs are rescaled in fitting the $b$ parameters of the transfer function within the DBM calibration process.  A transfer function model is not subject to a direct mass balance constraint, for example in flood forecasting applications where rainfall may be modelled against stage rather than discharge (e.g. Leedal et al., 2013).  A simple antecedent precipitation index (API) was also tried initially, although this introduces additional parameterisation; it worked with reasonable success for Newby Beck but not for the other catchments, and therefore, as a consistent method was sought for all catchments, the API approach was not pursued in this case.  For annual TP loads, the models (still with hourly timestep) were identified based on the data for hydrological years 2011/12 and 2012/13 for Newby Beck, but, because of missing output data, just for hydrological year 2012/13 for the Blackwater and Wylye catchments. Models were validated on the data for all, or part, of the hydrological year 2013/14.

Model fit was assessed according to model bias, to evaluate systematic over- or under-prediction of the model, and to $R_t^2$ (also known as Nash Sutcliffe Efficiency, NSE):

$$R_t^2 = 1 - \frac{\hat{\sigma}^2}{\sigma_{\hat{y}}^2} \tag{7}$$

where $\hat{\sigma}^2 = \frac{1}{N}\sum_{i=1}^{N}[y_i - \hat{y}]^2$ ; $\sigma_y^2 = \frac{1}{N}\sum_{i=1}^{N}[y_i - \bar{y}]^2$ ; $\bar{y} = \frac{1}{N}\sum_{i=1}^{N} y_i$ (8)

$\hat{y}$ is the model simulation; $\hat{\sigma}^2$ is the mean squared error of the model residuals (only equal to the variance if the mean of the residuals is zero) and $\sigma_y^2$ is the variance of the observations, $y_{i.}$. A balance of model fit and over-parameterisation was sought using the Young Information Criterion (YIC) and visual inspection of the model fit to the monitoring data. Model assessment criteria are defined in SI Section S2.

### 2.4 Uncertainty estimation

### 2.4.1 Structural uncertainty

The DBM technique involves the simplified representation of complex systems, based on the information in the data (Young, 1998; Young, 2001; Young et al., 2004). In practice, this means identifying models over a range of orders, and choosing the most appropriate model order. Generally the simplest (lowest order) model which balances model fit without over-parameterisation is chosen. The chosen models often have a very simple structure, which will certainly not be a true

representation of all the processes, but may model the data adequately. This structural error is accepted as part of the DBM technique, in order to reveal the dominant modes of response.

### 2.4.2 Parameter uncertainty

The Instrumental Variable algorithms, (RIVCBJ and RIVBJ), allow unbiased estimation of the model parameters and their covariance matrices. Monte Carlo sampling within the parameter space determined by the covariance matrices allows for

uncertainty in derived quantities, such as time constants, to be calculated. In general with DBM modelling, very little of the total uncertainty is due to the parameters, partly because there are so few of them and because the linear-dynamic part of the process that the model describes is well-defined. Note that in the case of transfer function models of the hydrograph, the models do not directly reflect the transport of water in the system since the hydrograph represents the integrated effects of celerities in the system rather than flow velocities (McDonnell and Beven, 2014).

### 2.4.3 Data uncertainty

A review of measurement data uncertainty is presented by McMillan et al., (2012), including uncertainties in rainfall observations. For all three catchments in this study, input data (rainfall) was based on three rain gauges in or near each catchment. This only gives a catchment rainfall estimate, which is affected by the non-homogeneity of the rainfall field and

the rainfall regime, and therefore some of the mismatch between model fit and observations (for any modelling technique) may be attributed to uncertainties in the rainfall input.

A rigorous treatment of the uncertainties in high frequency nutrient data and its subsequent impact on loads is given by Lloyd et al., (2016b). For Newby Beck, where stage-discharge gaugings were available, the discharge uncertainty was estimated using the method of McMillan and Westerberg (2015), fitting multiple plausible rating curves and weighting with a likelihood function. This method accounts for a mix of systematic and random measurement errors. The uncertainty on the phosphorus concentration measurements was estimated by comparing the time series from the bank-side analyser with the laboratory spot samples taken for ground-truthing (Lloyd et al., 2016b), fitting multiple regression curves and weightings according to McMillan and Westerberg (2015). The time series of discharge and TP concentration, with their uncertainty distributions were then combined by resampling to give the measurement data uncertainties on the TP loads. For the Wylye, discharge measurement uncertainties were estimated using a standard deviation of 10%, the maximum value calculated by Lloyd et al. (2016b) for the gauging site at Brixton Deverill using the method of Coxon et al. (2015). Wylye discharges were combined with a standard deviation of 0.11 mg L$^{-1}$ for the uncertainty on the TP concentration from the bank-side analysers (Lloyd et al., 2016b) to give uncertainty bounds on the TP load. For the Blackwater, discharge uncertainties were estimated by the DTC team and supplied with the DTC data, with uncertainty bounds of approximately $\pm$ 20% for low flows rising to $\pm$ 30% for high flows. This was combined with a standard deviation of 0.01 mg L$^{-1}$ for the uncertainty on the TP concentration from the bank-side analysers (Outram et al., 2016). Measurement data uncertainty bounds are shown on plots as a blue shaded band.

**3 Results and Discussion**

**3.1 Observed hydrological response and total phosphorus load in the three catchments**

Time series data from each catchment (Fig. 2) indicated large contrasts in the hydrological response of each study catchment, with Newby Beck (Eden) showing a very flashy response to rainfall (Fig. 2a). Although a fast response at certain times was also evident in the Blackwater (Wensum) catchment (Fig. 2c) and the Wylye (Avon) catchment (Fig. 2e), there was also a more pronounced seasonal response, particularly in the Wylye where a large groundwater component could be observed in the winter periods. This indicates the importance of both high-frequency data and a long-term record, to capture both fast and slower dynamics adequately. The errors resulting from sampling well below the catchment dynamics have been well documented elsewhere, (e.g. Johnes, 2007; Jones et al., 2012; Lloyd et al., 2016b; Moatar et al., 2013). TP concentrations in all three study catchments revealed peaks that corresponded with runoff, with maximum values of 1.0 mg L$^{-1}$, 0.9 mg L$^{-1}$ and 1.5 mg L$^{-1}$ in the Newby Beck, Blackwater and Wylye catchments, respectively. Newby Beck showed a very low background concentration of TP at low flow (minimum < 0.01 mg L$^{-1}$), compared to 0.05 – 0.1 mg L$^{-1}$ in the Blackwater, and around 0.12 mg L$^{-1}$ in the Wylye. The relationships between streamflow and TP concentration are shown in SI Figs S1 – S3,

and the relationships between streamflow and TP load are shown in SI Figs S4 – S6. The presence of a measurable, background, non-rainfall dependent concentration suggests an additional source of phosphorus to the recently applied agricultural sources. Such non-rainfall dependent sources include legacy stores of agricultural P in the soil, both large and smaller point source discharges, such as sewage treatment works and domestic septic tanks (Zhang et al., 2014), and groundwater, specifically contributions from mineral sources in the Upper Greensand geology of the Hampshire Avon (Allen et al., 2014).

A summary of the observed total rainfall, runoff, mean concentration and TP load is given in Table 1 for the period 1 October 2012 – 30 September 2013 (the hydrological year with the most complete dataset). The lowest mean annual TP concentrations were observed in the Newby Beck catchment, but combined with the highest runoff this resulted in a high total annual TP load. Conversely, although mean annual TP concentration in the Blackwater was also higher than in Newby Beck, when combined with the lowest runoff, this resulted in the lowest total annual TP load. The rainfall-runoff ratio for Newby Beck (0.65) was much higher than for the Blackwater (0.31) or the Wylye (0.32), indicating a larger capacity for storage in the latter two catchments. Despite similarity in the rainfall-runoff ratio, total runoff in the Wylye was higher than the Blackwater because of the higher total rainfall.

Detailed analysis of the high-frequency data is not included here as it has already been published by several authors (e.g. Ockenden et al., 2016; Outram et al., 2014 (including hysteresis analysis); Perks et al., 2015). Investigation of the relationships between TP concentration and streamflow indicated that, for all three catchments, the TP concentration was out of phase with the streamflow; distinct hysteresis loops (SI Figs S1 – S3), also observed by Outram et al. (2014), showed different TP concentrations on the rising stage of a storm hydrograph compared to the same stage on the falling hydrograph. This indicates that antecedent conditions and the storage state of the catchment are important in determining the response. In order to capture the effects of storage, dynamic models are required.

**3.2 Identification of linear transfer function models for short storm sequences**

For short storm sequences up to about a month, when antecedent flows for events were rather similar, linear models were identified for the Newby Beck catchment. These were useful for infilling missing discharge or TP load data, or for highlighting and estimating uncertainties in discharge and TP load when extrapolation of the stage-discharge relationship was inappropriate. The model is only reliable for the conditions covered during the calibration period, but it may still be useful when there are known problems with a stage-discharge relationship (such as during extreme events). Indeed, the stage to discharge relationship is the weakest point of all the catchment models relying on stage measurements. Whilst it was still possible to identify linear models for short periods for the Blackwater and Wylye catchments, the parameter uncertainty for these models was large; the parameters cannot be well constrained when the (slow) time constant was of similar order to the

period of identification. For this reason, linear models for short periods for the Blackwater and the Wylye were not considered useful.

Table 2 shows results from rainfall-runoff and rainfall-TP load models identified for Newby Beck for a series of contiguous storms in November 2015, immediately preceding Storm Desmond (5 – 6 December 2015), which caused catastrophic flooding in Cumbria and Lancashire, UK. During Storm Desmond, Honister Pass in Cumbria received the highest 24 h rainfall on record (341 mm) and Thirlmere received the highest 48 h rainfall on record (405 mm). The storm was remarkable for the duration of sustained rainfall. At Newby Beck, 156 mm of rainfall was recorded in 36 h. Although the monitoring equipment was recording during Storm Desmond, the peak flows during the storm were out of bank for around 31 h (compared to less than 3.5 h during more typical storms), with anecdotal evidence that the gauging point was significantly bypassed, so these out of bank flows were highly uncertain. This measurement uncertainty is shown by the shaded bands in Fig. 3 (discharge model) and Fig. 4 (TP load model), which span the observed (calculated from stage) discharge and TP load. This is more visible in the zoomed-in periods for discharge (Fig. 3b) and TP load (Fig. 4b). Concentrations were assumed to be reasonably accurate, but TP loads were underestimated due to the underestimate of discharge. Storm Desmond was not included in the model identification period. Using the models from the November period to simulate flows (Fig. 3) and TP load during Storm Desmond (Fig. 4) suggests that both discharge and TP load were underestimated. Time series and histograms of the residuals are given in SI Fig. S7 for discharge and SI Fig. S8 for TP load. The zoomed-in period for the TP load model (Fig. 4b) suggests that whilst the transfer function model got the timing of the load peak and the decay approximately right, the model generally started to respond before the observed load responded.

Although there are uncertainties associated with whether it is valid to extend the models identified above to an extreme event such as Storm Desmond, we believe that this highlights the possible underestimation in discharge and TP load during Storm Desmond and that the models in Table 2 might provide more realistic estimations of the true values.

**3.3 Identification of transfer function models on annual time series data**

Longer term models, based on two years of hourly data, were identified for each catchment. Model fits ($R_t^2$) for rainfall-runoff models for the identification period (Table 3) were 0.71 for Newby Beck and 0.87 for Wylye, but only 0.37 for the Blackwater. Model bias was less than ± 10% for all three catchments. The runoff models were all linear transfer function models relating effective rainfall to discharge, where the exponent in the non-linear relationship between rainfall and effective rainfall (Eq. 6) was optimised at the same time as model parameter identification. The non-linearity, which reflects the effect of the antecedent soil moisture conditions in the catchments, was accounted for with the soil moisture surrogate expressed as a power function of discharge (Beven, 2012) with exponent β in Eq. 6, where a value of zero produces a linear response to rainfall and a higher value leads to an increasingly non-linear response. The β values identified for Newby Beck, Blackwater and Wylye were 0.37, 0.65 and 0.59, respectively, indicating the most non-linear response was in the Wensum

(Blackwater) catchment, which also gave the lowest model efficiency values. The best identified model for rainfall-runoff in each catchment was a second-order model. In general, models higher than second order gave little improvement in model fit but a large deterioration in YIC, signifying over-parameterisation not warranted by the information in the monitoring data, whereas first order models often gave a reasonable fit to the model peaks (and hence reasonable $R_t^2$), but poor fit to recession periods.

The dynamic response characteristics of time constant and percentage on each flow pathway (for definitions see SI Table S4), determined after partial fraction decomposition, can be compared between the study catchments for both discrete and continuous time models. The time constants are associated with the dominant pathways and indicate how quickly each impulse response (of water or TP mass) is depleted to 37% (or fraction 1/e) of the peak exported. This is the standard definition of a time constant in a first order linear time-invariant dynamic process e.g. $A(t) = A_0 \exp(-t/T_c)$ where $T_c$ is the time constant. In reality there will be a continuum of runoff pathways with different time constants (Kirchner et al., 2000), but the information in the data indicates that this continuum can be simplified by representation as just two dominant pathways.

The marginal distributions of the time constants and proportion of flow or TP load (Table 3) were determined from 1000 - 10,000 Monte Carlo realisations using the covariance of the parameter estimates. The parameter uncertainties estimated within the DBM methodology were small, even for the response characteristics of the TP load models, which had higher uncertainty than rainfall-runoff models; TP load models had coefficients of variation of less than 3% for fast time constants, less than 6% for slow time constants and less than 2% for proportions on pathways. For the rainfall-runoff models, the time constant for the fast pathway was 2.9 h ± 0.1 h for Newby Beck, with 43% ± 0.5% of the water taking this pathway; in the Wylye, the time constant for the fast pathway was 4.1 h ± 0.2 h, but with only 8% ± 0.2% of the water taking this route. This is consistent with the much higher baseflow index in the Hampshire Avon (0.93) than the Eden (0.39) (SI Table S1), which is clearly visible in the data (Fig. 1). For the Blackwater, 25% ± 0.6% of the flow took the fast pathway, which is also consistent with the baseflow index in the Wensum (0.8) being between the Eden and Hampshire Avon. The fast time constant for the Blackwater catchment was much slower, at 14.8 h ± 0.25 h; this may be related to the average slope of the catchment, which is much lower for the Blackwater catchment (less than 2%) compared to 6 − 8% for the Wylye and Newby Beck catchments. The slow time constant for Newby Beck was 147 h ± 5 h, with 57% ± 0.5% of flow taking this pathway; this compared with 441 ± 13 hours (75% ± 0.6 % of flow) for the Blackwater and 395 ± 6 hours (92% ± 0.2% of flow ) for the Wylye.

### 3.4 Interpretation of TP load dynamics alongside runoff dynamics

For the rainfall-TP load models, at Newby Beck the best identified model was a first order model relating the effective rainfall (from the runoff model, i.e. calculated one step at a time using the simulated discharge, Qsim) to the TP load (Table

3, Fig. 5). Although it was possible to identify a second order model, this made virtually no difference to model fit, $R_t^2$, and at the expense of YIC (signifying over-parameterisation), and decomposition of the model revealed time constants for the two pathways that were both less than 8 hours (c.f. 147 hours for the slow pathway for the rainfall-runoff model in Table 3). This indicates that in Newby Beck, all the TP load is transported through a quick flow pathway. This is consistent with most

of the load being associated with P mobilised from diffuse agricultural sources, which is transferred by surface runoff or shallow sub-surface flow. This includes particulate P transported in surface runoff or drain flow (Heathwaite et al., 2006), subsurface movement of fine particles and colloids (Heathwaite et al., 2005), and displacement of fast subsurface soluble P sources. Young (2010) recommended a minimum data sampling rate of one-sixth of the time constant, in order to avoid possible temporal aliasing effects. Littlewood and Croke (2013) illustrated the parameter inaccuracy and loss of data when

observations were under-sampled for discrete time transfer functions, with inaccuracy decreasing and parameter estimates approaching stable values as the sampling interval decreased from 24 hours (daily sampling) down to hourly sampling. The time constant for the first-order TP load model for Newby Beck was 1.6 ± 0.04 hours. In this study, daily data would not capture the true dynamics of discharge and TP load, and that, ideally, for flashy catchments such as Newby Beck, a sampling interval shorter than hourly would be even more robust. However, for the other catchments in this study, the hourly data

frequency was sufficient. The time constant for the TP load model (1.6 ± 0.04 h) was even faster than the fast time constant for the second-order (two pathway) rainfall-runoff model (2.9 h ± 0.1 h), indicating that the TP mass impulse response was depleted at a faster rate than the water response, i.e. that the store was diluted as the storms progressed or that the sources must be readily connected and closer to the stream, since TP depends on transport velocities and we would normally expect velocities to be less than celerities under wet and surface runoff conditions. Those source areas would also be the most

readily exhausted so the effects would reinforce each other.

Expanded sections of Fig. 5 are shown for storms in May 2012 (Fig. 6a) and November 2012 (Fig 6b). Time series of residuals and residuals against observed values are given for the discharge model in SI Fig. S9 and for the TP load model in SI Fig. S10. Although Fig. 5 illustrates several storms where the model underestimated the peak TP load, the model matched

the shape and peak of the May 2012 storm quite well. However, once again the model started to respond to the rainfall before the observations showed a response. Fig 6b shows an example of a storm in which the TP load was underestimated by the model. The model parameter uncertainty was considerably smaller than the measurement data uncertainty. The model did not always lie within the bands indicated by the measurement data uncertainty, whereas the total model prediction uncertainty (including the residual uncertainty) would span most of the observations, indicating that the simple structure of

the model does not capture all the dynamics, and that there are other sources of uncertainty (such as rainfall input) which are not quantified.

For the Wylye, the best identified TP load model was a second-order model relating effective rainfall to TP load, with 42% ± 1% on a fast pathway (TC = 6.1 ± 0.3 hours) and 58 ± 1% on a slower pathway (570 ± 54 hours) (Table 3, Fig. 7).

Compared to the runoff model, this showed a much greater percentage of the TP load on faster pathways such as surface runoff, shallow sub-surface flow or sub-surface drains. Nevertheless, there was still a significant proportion travelling on a slower pathway, which highlights the need for pollution mitigation efforts to include measures that take account of sub-surface and groundwater flows, and also, to recognise that surface runoff from farmland is not the only source of nutrients and sediment (Allen et al., 2014; Evans, 2012). These models cannot provide spatial information, but having identified that a slow pathway is so important, measures which prevent pollutants getting to the slow pathway in the first place, such as reductions at source, will be helpful. This may require further specific measurements, such as testing P in soils or identifying septic tanks in the catchment. With DBM models, this interpretation is made a posteriori, after the data assimilation and is based on inferences from the objectively identified dominant modes of the system response.

Fig. 8 shows expanded sections of the Wylye TP load model, including a large storm in which the load is underestimated (Fig. 8a) and two smaller storms where the model overestimated the loads (Fig. 8b). For the Wylye catchment, the measurement uncertainty was dominated by the uncertainty on the data from the TP sensor, rather than the uncertainty in the discharge (Lloyd et al., 2016b). However, some of the mismatch between model and observations here might also be attributable to uncertainty in rainfall input: in Fig. 8a there could be an underestimate in catchment rainfall not captured by the rain gauges; conversely, in Fig. 8b the rain gauges may have captured more than the catchment average rainfall. Time series of residuals and residuals against observed values are given for the Wylye discharge model in SI Fig. S11 and for the TP load model in SI Fig. S12.

The TP load model used for the Blackwater was a linear model relating rainfall directly to TP load. The second-order TP model gave fast and slow time constants of $12.5 \pm 0.6$ hours and $376 \pm 44$ hours, respectively (Table 3, Fig. 9). The time constants were similar in magnitude, though both slightly shorter, to the time constants for the runoff model, suggesting a possible exhaustion effect where, as in Newby Beck, the TP mass store was diluted as the response progressed. For the Blackwater, as in the other study catchments, the proportion of TP load transferred on the fast pathway ($54 \pm 2\%$) was considerably more than the proportion of water on the fast pathway ($25\% \pm 0.6\%$). Although seasonal non-linearity was still evident in the data from Blackwater, the rainfall-runoff models that included the non-linearity did not validate very well (SI Fig S18), such that the two-stage TP models using the effective rainfall calculated one step at a time using the simulated discharge, Qsim, gave a worse fit to the data than a simple linear model. This may have been due to missing data in the discharge and TP time series, particularly over the storm peaks or to inadequate representation of P inputs. An expanded section of Fig. 9, showing a series of storms in December 2012 (Fig. 10a) indicates the seasonal non-linearity of the response, which cannot be captured with a linear model, with a linear rainfall input. The first storm was considerably underestimated, but later storms were overestimated. This can usually be accounted for by using a non-linear effective rainfall input, which was unsuccessful in this case. A storm in May 2013 (Fig. 10b), when the land might have been drier than during the December storms, showed considerable overestimation of TP load by the linear model fitted to the December

period. Time series of residuals and residuals against observed values are given for the Blackwater discharge model in SI Fig. S13 and for the Blackwater TP load model in SI Fig. S14.

The proportion of TP load exported on the fast pathway was considerably greater for all catchments than the corresponding proportion of water on the fast pathway, by a factor of approximately two for Newby Beck and Blackwater and approximately five for the Wylye. This suggests that on the fast water pathways, generally associated with shallower pathways such as shallow sub-surface flow, field drains and surface runoff, there is more release of TP than on deeper water pathways. This is consistent with soil profiles in agricultural areas, which generally show P concentrated on the surface and in the near-surface soil layers, with a decrease in P with depth (Heathwaite and Dils, 2000).

Validation of the TP model for Blackwater and Wylye was performed on a shorter period than for Newby Beck (half of the hydrological year 2013/14) because of missing data (Table 3, SI Figs. S15-S18). The power law used to represent the rainfall-runoff non-linearity did not validate very well in the Blackwater catchment. Different representations of the rainfall-runoff linearity were also investigated, such as the Bedford Ouse Sub-Model (Chappell et al., 2006; Young, 2001; Young and Whitehead, 1977), in which the soil storage is related to an antecedent precipitation index. Although changes in the model non-linearity representation made minor differences to model fit, none of the model variants validated well for the Blackwater catchment. This suggests that there may be a different mechanism at work in the Blackwater catchment, in which a fast pathway only becomes active once the soil is fully saturated, or the groundwater level rises to a certain level (Outram et al., 2016). This could be due to the shallow slopes, which encourage infiltration rather than runoff. Alternatively, the response may be more dominated by point sources which are not as rainfall-driven, or sources such as sediment-laden runoff from impervious surfaces (roads/yards), which are rainfall-driven but do not behave in the same non-linear way as the runoff from soil.

In addition, the conditions experienced during the two years used for model identification may not be very similar to the validation period. From the data in Fig. 1c, the winter of 2011 and spring of 2012 showed much lower discharge than the same months in subsequent years. The groundwater recharge, which is shown as an increase in the baseflow in winter, was obvious for winter 2012/13 and winter 2013/14 for both the Blackwater (Fig. 2c) and the Wylye (Fig. 2e), but was not evident for either catchment for the winter of 2011/12. Because of the slow time constants for these catchments, the dataset for model identification needs ideally to be longer than for the Newby Beck catchment, where the dynamics are much faster. This study suggests that the dataset used here was not long enough for the Blackwater catchment to capture an adequate range of conditions.

## 3.5 Advantages and limitations of the modelling method

The benefits and limitations of the modelling method for TP load are summarised in Table 4.  For catchments that exhibit rapidly changing dynamics, such as response to storm events, models calibrated with daily data will have large uncertainties associated with the parameters (and output) because the input data do not capture the high frequency dynamics of processes such as P transfer.  This study shows that simple transfer function models using data with sub-daily resolution can simulate the dynamics of TP load, with model fits at least as good as generally achieved with process-based models (Gassman et al., 2007; Moriasi et al., 2007) and with low parameter uncertainty.  Full direct model comparisons are not currently possible, as the published results for process-based models used different catchments and data sets.  It is still advisable to validate a fitted model using at least a split record test (Klemes, 1986).  This highlights the importance of long and complete datasets with good time resolution for properly representing both flow and TP loads for such catchments.  The high data demand of DBM models is noted in Table 4. Technology and monitoring methods are improving all the time so that high-frequency data are now more readily available  (e.g. Jordan et al., 2007; Jordan et al., 2005; Outram et al., 2014; Skeffington et al., 2015) This requirement for adequate datasets is often an obstacle in the use of the DBM modelling method, but as such datasets become more available, the method can be used to improve our understanding of catchments.  We should embrace efforts to improve data coverage and ways to use it widely.

The models in Table 3 have been identified using a consistent method, , to test how well this modelling method copes with the different characteristics of the three catchments.  The method has been successfully applied to all the catchments, although less successfully for the Blackwater catchment.  It is likely that the models could be improved if catchment-specific adjustments were made or used alongside other models in a hypothetico-inductive manner (Young, 2013).  For instance, in the Blackwater catchment, the use of state dependent parameters (Young, 1984) might be more successful to capture the rainfall-runoff non-linearity. This means that, rather than using the form of the non-linearity specified by Eq. 6, the parameters could be allowed to vary according to some other observed state.  In addition, model fit might be improved by accounting for heteroscedasticity of residuals (shown in residual analysis, SI Figs. S9-S14), through transformation of data and residuals (e.g. Yang et al., 2007). Models for all catchments could be improved by having a longer dataset, to ensure, as far as possible, that environmental conditions during a future simulation period have already been experienced during the identification period.

The models showed a pattern of underestimation of high-level TP load events and, to a lesser extent, overestimation of lower level events, (SI Figs 10, 12 and 14).   This was more apparent for TP load than for the discharge model (SI Figs 9, 11 and 13), although in many cases this was within the limits of the uncertainty in the observed data.   This suggests that, for the TP load model, the non-linearity may be rainfall, discharge or load-dependent to a greater extent than allowed for in the nonlinearity of Eq. 6. This could be explored using State Dependent Parameter estimation, on which the power law of Eq, 6

for the flow nonlinearity was originally based ( Young and Beven, 1994; Young, 1984). In addition, models with at least two terms in the numerator polynomial could provide more flexibility for a differencing effect, i.e. a consistent flushing effect with higher load occurring during the rising limb of the discharge peak. This mechanism is not represented in first order models [1 1 del], as for Newby Beck, as it requires two terms of the numerator polynomial.

The use of process-based models is often justified on the basis that the inclusion of adequate process representations will lead to more robust estimation of the response to changing environmental conditions. This is the basis for arguing that process-based models are better suited for predicting the impacts of future change. However, they also involve a plethora of (often difficult to validate) assumptions in their model structures and parameters. In practice, parameters set during

calibration are rarely changed to account for changes in the modelled processes under future conditions, although by calibrating models for conditions similar to the expected future conditions, it may be possible to incorporate non-stationary parameter values (Nijzink et al., 2016). This idea could be integrated into DBM models by choosing identification periods which are most likely to reflect the conditions of the simulation period or through the use of state-dependent parameters. Thus, whilst the data-based assumption of similar conditions may be questioned when limited periods have been used for

identification, usually restricted by data availability, we argue that many of the factors contributing to catchment response will not have changed (e.g. catchment topography, soil type and geology) and that this assumption will in many circumstances be no more restrictive than the (different) assumptions made when using process-based models. Clearly, where the factors contributing to catchment response have obviously changed (such as if all septic tanks were upgraded or if farm budgeting reduced the additions of P), then simple transfer function models would not be able to predict the changes

over time, whereas, in theory, process-based models might be able to account for such changes, albeit with much uncertainty, (e.g. Dean et al., 2009; Yang et al., 2008). However, for rainfall dominated responses, or responses to changes in rainfall patterns, simple transfer function models can provide valuable understanding of the dominant modes of a catchment, which, in turn, can be used to target management interventions.

**4 Summary and Conclusions**

High temporal resolution data (hourly) of discharge and TP load have been used to identify simple transfer function models that capture the dynamics of rainfall-runoff and rainfall-phosphorus load in three diverse agricultural catchments. Linear models were identified for short storm sequences in the flashy Newby Beck catchment, when antecedent flows for events were similar. Models identified for November 2015 were used to simulate flows and TP loads in the devastating Storm

Desmond (5-6 December 2015), supporting our belief that the discharge and TP load calculated from recorded data during this storm were considerably underestimated. In these circumstances, simple models could be useful to infill missing data or to highlight or estimate uncertainties in the recorded data. Linear models for short periods were not appropriate for the less

flashy Blackwater and Wylye catchments when the slow time constant (for a second order model) was similar in length to the time period of identification, making the parameter uncertainty large.

Longer-term models were identified for each of the three catchments on two years of data. Comparison of rainfall-runoff and rainfall-TP load models for each catchment allowed a better understanding of the dominant modes of transport within each catchment, which was based on the times series data alone, rather than other (unmeasured) catchment parameters. In all three catchments, a higher proportion of the TP load was exported via a fast pathway than the corresponding proportion of water on the fast pathway. In agreement with soil profiles in agricultural areas, this suggested that there is more release of TP on fast (generally shallower) water pathways such as shallow sub-surface flow, field drains and surface runoff.

For successful simulations of future conditions, the models require long datasets to ensure that a full range of driving conditions has been included in the identification period. However, this study shows that simple transfer function models can be successful in modelling TP loads and explaining dominant transport modes. Transfer function models make good use of high frequency data, require very few parameters with low uncertainty and allow physical interpretation based solely on the information in the data.

Data availability

The data used in this study are openly available from Lancaster University data archive at https://dx.doi.org/10.17635/lancaster/researchdata/ (reserved until publication).

Supporting Information

Estimation of hourly rainfall time series for the Wylye catchment (Section S1); Model assessment criteria (Section S2); Study catchment characteristics (Table S1); Notation (Table S2); Structure of models and relationship between parameters from discrete-time and continuous-time models (Table S3); Definition of time constants, steady state gains and fraction on each pathway for discrete-time and continuous-time models (Table S4); Model structures and parameters identified (Table S5); Hourly streamflow against total phosphorus concentration for the Newby Beck catchment (Fig. S1), the Blackwater catchment (Fig. S2) and the Wylye catchment (Fig. S3); Hourly streamflow against total phosphorus load for the Newby Beck catchment (Fig. S4), the Blackwater catchment (Fig. S5) and the Wylye catchment (Fig. S6); Time series of residuals and histograms of residuals for short term model, Newby Beck (Figs. S7-S8); Residual analysis, long-term models (Figs. S9-S14); Model validation (Figs. S15-S18). This material is available online.

Author Contributions

M.C.O. ran the DBM model and led the writing of the paper. W.T. assisted with DBM modelling. P.M.H was overall project lead with K.J.B., P.W., P.D.F. and J.Z. also helping manage the project. All authors participated in interpretation of results and the writing and editing process. M.C.O., K.J.B., A.L.C., R.E., P.D.F.., K.J.F., K.M.H., M.J.H., R.K., C.J.A.M., M.L.V., C.W., P.J.W., J.G.Z. and P.M.H. contributed to NUTCAT 2050; A.L.C., K.M.H., S.B., R.J.C., J.E.F. and P.M.H. are part of

5   the DTC project.

Competing interests

Jim Freer is a member of the editorial board of Hydrology and Earth System Sciences.

Acknowledgements

This work was funded by the Natural Environment Research Council (NERC) as part of the NUTCAT 2050 project, grants NE/K002392/1, NE/K002430/1 and NE/K002406/1, and supported by the Joint UK BEIS/Defra Met Office Hadley Centre

Climate Programme (GA01101). The authors are grateful to the UK Demonstration Test Catchment (DTC) research platform for provision of the field data (Defra projects WQ02010, WQ0211, WQ0212 and LM0304). The DTC data are available at http://www.environmentdata.org/dtc-archive-project/dtc-archive-project .

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

**Table 1  Observed rainfall, discharge, total phosphorus (TP) concentration and load for the period 1 October 2012 -30 September 2013, for the three catchments**

| Catchment | Total rainfall (mm) | Total runoff (mm) | Rainfall-runoff ratio | % discharge data missing | Mean annual discharge $(m^3\ s^{-1})$ | Mean annual TPconc $(mg\ L^{-1})$ | Total annual TPload (kg) | % TPload data missing |
|---|---|---|---|---|---|---|---|---|
| Newby Beck Eden, Cumbria | 1186 | 776 | 0.65 | 0.0 | 0.31 | 0.080 | 1577 | 19.7 |
| Blackwater, Wensum, Norfolk | 634 | 195 | 0.31 | 13.8 | 0.14 | 0.092 | 277 | 30.6 |
| Wylye, Avon, Hampshire | 850 | 273 | 0.32 | 0.3 | 0.44 | 0.149 | 1705 | 27.4 |

**Table 2** Rainfall-runoff and rainfall-total phosphorus load (TP) models identified for Newby Beck during the period 7 November – 4 December 2015, with estimations of discharge and TP load during Storm Desmond (5/6 December 2015). CT linear = Continuous-time transfer function with linear rainfall input; $R_t^2$ = model efficiency measure (Eq. 7); $TC_{fast/slow}$ = time constant for the fast/slow pathway; $\%_{fast/slow}$ = percentage of output taking the fast/slow pathway; Model bias = $100 * \Sigma(y_i^{model} - y_i^{obs})/ \Sigma(y_i^{obs})$;

| Model | Model structure | $R_t^2$ | $TC_{fast}$ (h) | $TC_{slow}$ (h) | $\%_{fast}$ | $\%_{slow}$ | Model bias % | Σobs during Desmond | Σmodel during Desmond | % diff |
|---|---|---|---|---|---|---|---|---|---|---|
| Rainfall-runoff | CT linear [2, 2, 1] | 0.91 | 3.6 ± 0.4 | 33 ± 8 | 55 ± 5 | 45 ± 5 | 0.7% | 86.6 mm | 106.5 mm | 23% |
| Rainfall-TP load | CT linear [1, 1, 1] | 0.74 | 2.7 ± 0.3 | | 100 | | 13% | 196.5 kg | 273.6 kg | 39% |

**Table 3 Structure, response characteristics and model fit statistics of rainfall-runoff and rainfall-TP load models for each catchment. Models were calibrated on all or part of hydrological years 2012 and 2013 and validated on all or part of hydrological year 2014. $\beta$ = exponent in the power law used for rainfall-runoff non-linearity (Eq. 6); $R_t^2$ = model efficiency measure (Eq. 7); Qobs = observed discharge; Qsim = simulated discharge, using only the rainfall input; Model bias = $100 * \Sigma(y_i^{model} - y_i^{obs}) / \Sigma(y_i^{obs})$; TC$_{fast/slow}$ = time constant for the fast/slow pathway; %$_{fast/slow}$ = percentage of output taking the fast/slow pathway;**

| Location | Time period (calib) | Model | Model structure | $\beta$ | $R_t^2$ for calib (using Qobs) | $R_t^2$ for calib (using Qsim) | Model bias (calib) % | TC$_{fast}$ (h) | TC$_{slow}$ (h) | %$_{fast}$ | %$_{slow}$ | Time period (valid) | $R_t^2$ for valid (using Qsim) | Model bias (valid) % |
|---|---|---|---|---|---|---|---|---|---|---|---|---|---|---|
| Newby | 1.10.11 to 30.9.13 | R-Re-Q | CT [2, 2, 1] | 0.37 | 0.86 | 0.71 | -9.7 | 2.9 ± 0.1 | 147 ± 5 | 43 ± 0.5 | 57 ± 0.5 | 1.10.13 to 30.9.14 | 0.78 | -14.3 |
| Newby | 1.10.11 to 30.9.13 | R-Re – TPload* | CT [1, 1, 1] | | | 0.69 | 2.3 | 1.6 ± 0.04 | | 100 | | 1.10.13 to 30.9.14 | 0.62 | 5.1 |
| Blackwater | 1.12.11 to 31.8.13 | R-Re-Q | DT [2, 2, 6] | 0.65 | 0.82 | 0.37 | -1.5 | 14.8 ± 0.5 | 441 ± 13 | 25 ± 0.6 | 75 ± 0.6 | 1.10.13 to 30.9.14 | 0.32 | -9.4 |
| Blackwater | 26.10.12 to 28.7.13 | R - TPload | CT [2, 2, 4] | | | 0.67 | 5.4 | 12.5 ± 0.6 | 376 ± 44 | 54 ± 2 | 46 ± 2 | 1.10.13 to 31.3.14 | 0.31 | 38.2 |
| Wylye | 1.10.12 to 30.9.13 | R-Re-Q | DT [2, 2, 6] | 0.59 | 0.94 | 0.87 | 3.0 | 4.1 ± 0.2 | 395 ± 6 | 8 ± 0.2 | 92 ± 0.2 | 1.12.13 to 20.5.14 | 0.79 | 11.0 |
| Wylye | 1.10.12 to 30.9.13 | R-Re – TPload* | CT [2, 2, 6] | | | 0.67 | 5.5 | 6.1 ± 0.3 | 570 ± 54 | 42 ± 1 | 58 ± 1 | 1.12.13 to 31.3.14 | 0.50 | -19.7 |

*The effective rainfall – TPload model is a two-stage model;  it is assumed that the discharge is unknown, so that the effective rainfall must be calculated one step at a time, as Qsim is generated with the previously identified parameters of the rainfall-discharge model.  Hence Rt2 using Qobs is a one-step ahead prediction, whereas Rt2 using Qsim is a true simulation, only using the rainfall input.

**Table 4  Advantages and limitations of the DBM modelling method for rainfall-TP load**

| Advantages | Limitations |
|---|---|
| No prior assumption of model structure required | Requires complete, high temporal frequency datasets |
| Very few parameters required | Requires long datasets to cover a full range of driving conditions |
| Low parameter uncertainty | Models may not work well for future conditions if the range of conditions has not been included in the identification period |
| Makes good use of high frequency data | The power law to represent the rainfall-runoff non-linearity may not be the best representation for each catchment |
| Physical interpretation is made based only on the information in the data | Stationary DBM model will not capture time variable gains |

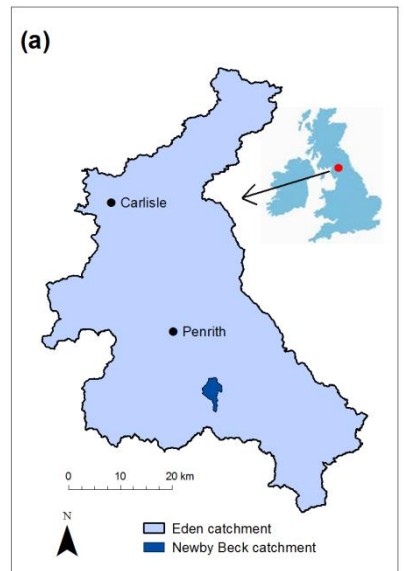
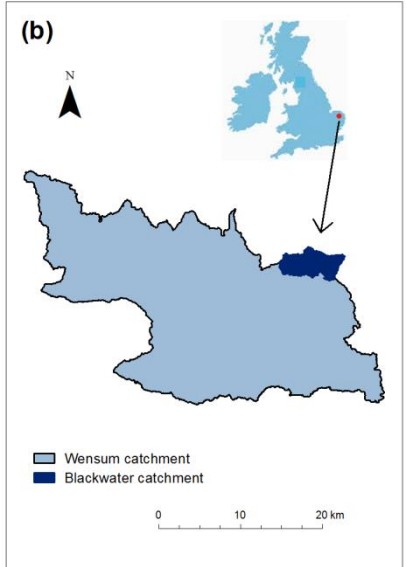
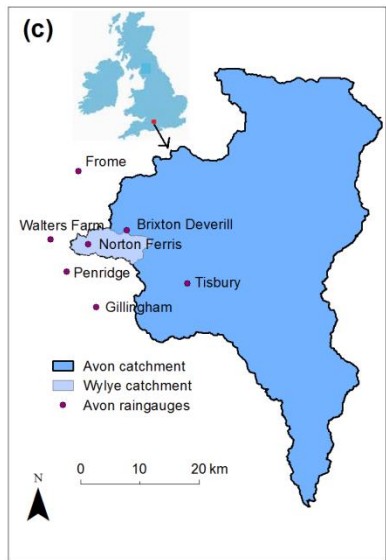

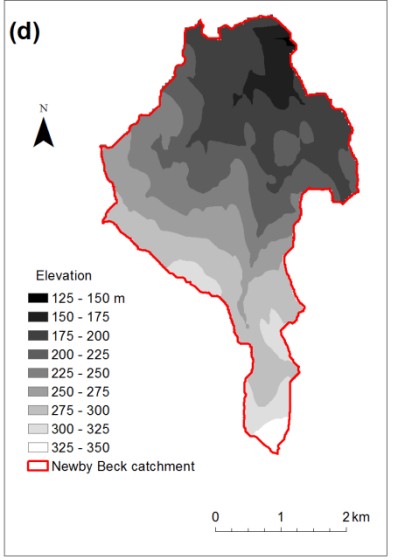
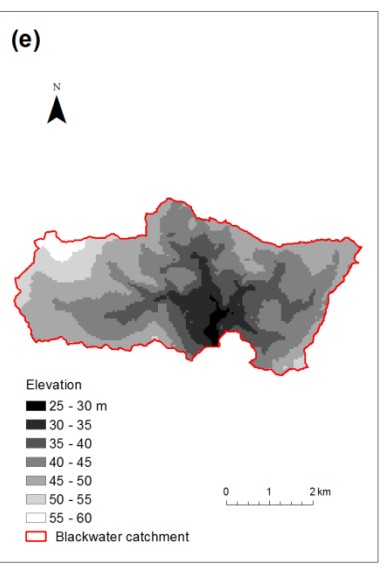
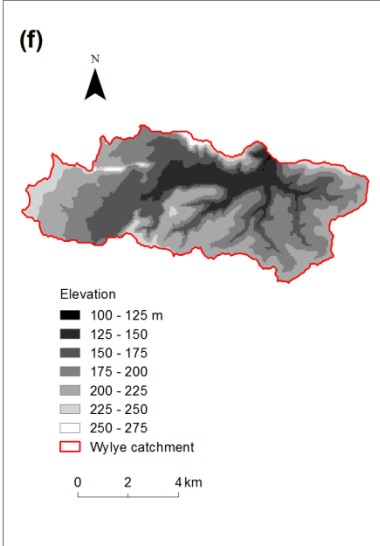

**Figure 1 Location and topography of study catchments. Newby Beck, Eden, Cumbria: location (a) and topography(d); Blackwater, Wensum, Norfolk: location (b) and topography (e); Wylye, Avon, Hampshire: location (c) and topography(f). © OS Terrain 50 DTM [ASC geospatial data], Scale 1:50000, Tiles: ny51, ny52, ny61, ny62, Updated: July 2013; Tiles st73, st83, tg02, tg12, Updated: 2 August 2016; Ordnance Survey (GB), Using: EDINA Digimap Ordnance Survey Service, http://digimap.edina.ac.uk; Downloaded: 2017-01-03.**

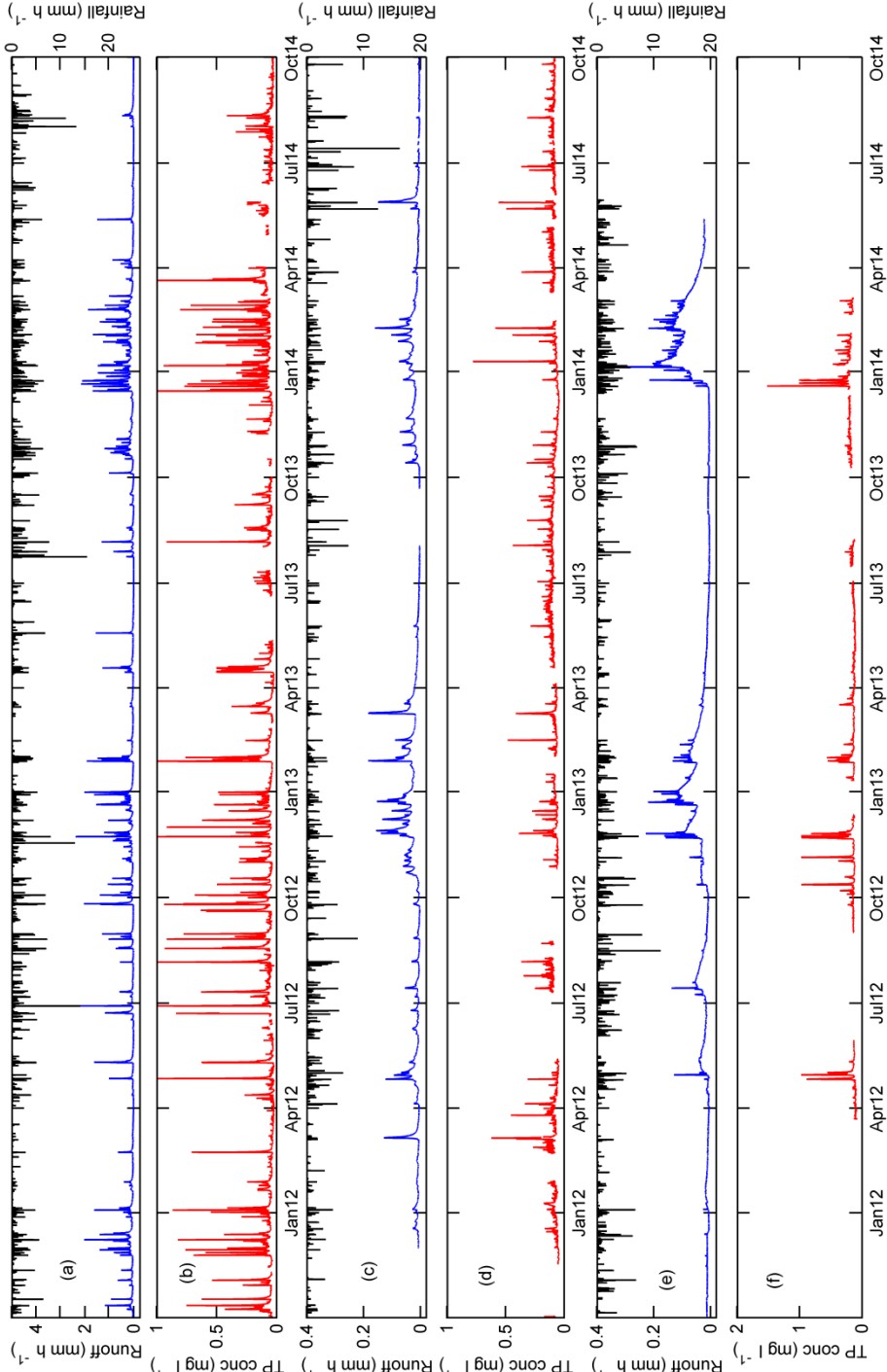

**Figure 2   Time series of hourly rainfall, runoff and total phosphorus (TP) concentration at the three Demonstration Test Catchments; rainfall and runoff (a) and TP concentration (b) at Newby Beck, Eden; rainfall and runoff (c) and TP concentration (d) at Park Farm, Blackwater, Wensum; rainfall and runoff (e) and TP concentration (f) at Brixton Deverill, Wylye, Avon.**

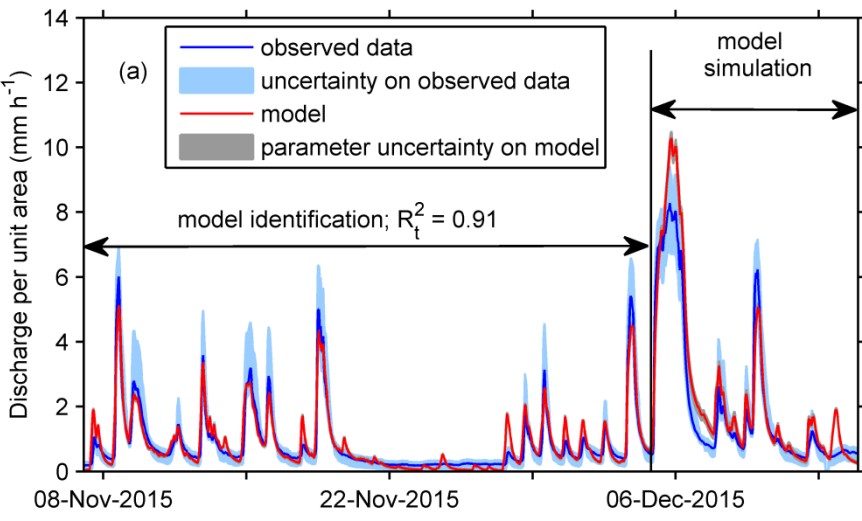

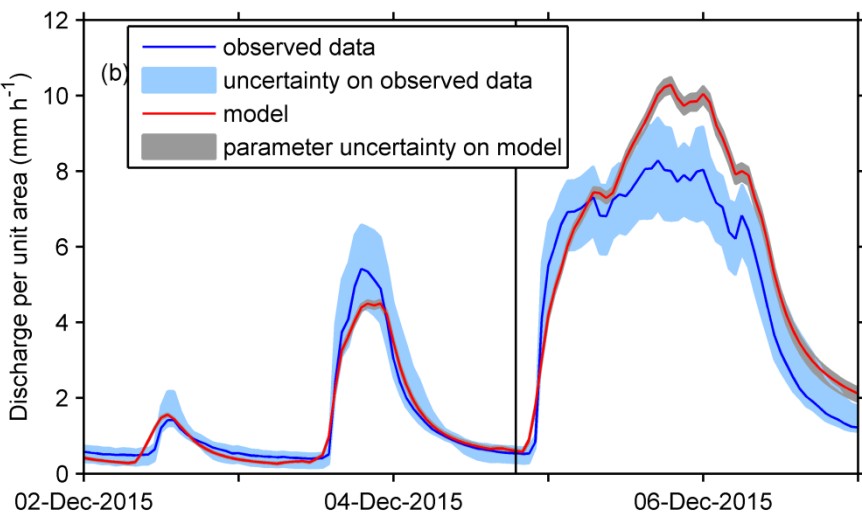

**Figure 3** Observed and modelled discharge per unit area **(a)** and zoomed section of the same **(b)** in Newby Beck, Eden during November 2015, with the same model used to estimate discharge during Storm Desmond 5/6[th] December 2015. The blue band indicates the 95% uncertainty bounds on the measurement data and the grey band indicates the 95% confidence limits on the parameter uncertainty. Total model predictive uncertainty (including the residual uncertainty) is larger than parametric uncertainty and would enclose the observations most of the time.

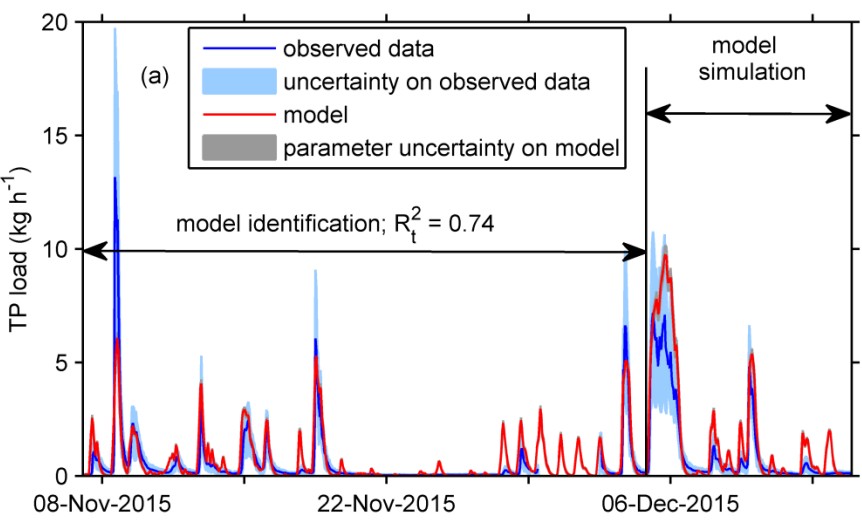

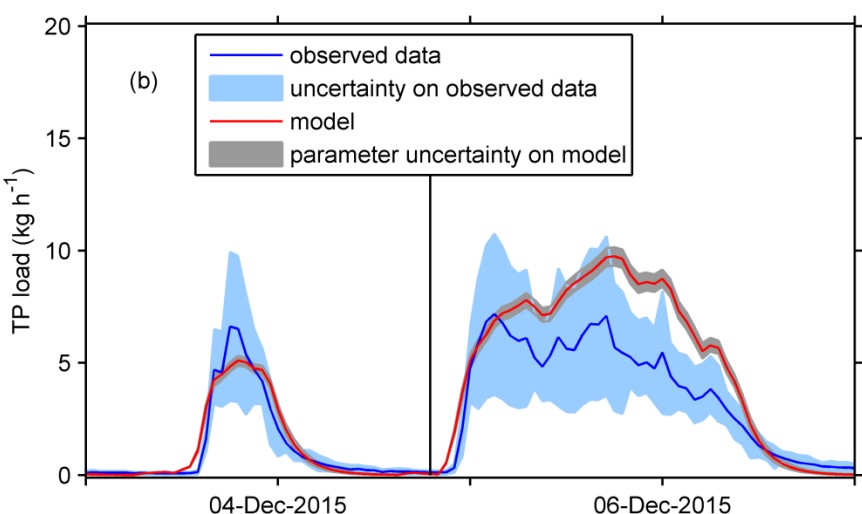

**Figure 4  Observed and modelled total phosphorus (TP) load (a) and zoomed section of the same (b) in Newby Beck, Eden during November 2015, with the same model used to estimate TP load during Storm Desmond 5/6th December 2015.  The blue band indicates the 95% uncertainty bounds on the measurement data.  The grey band indicates the 95% confidence limits on the parameter uncertainty.  Total model predictive uncertainty (including the residual uncertainty) is larger than parametric uncertainty and would enclose the observations most of the time.**

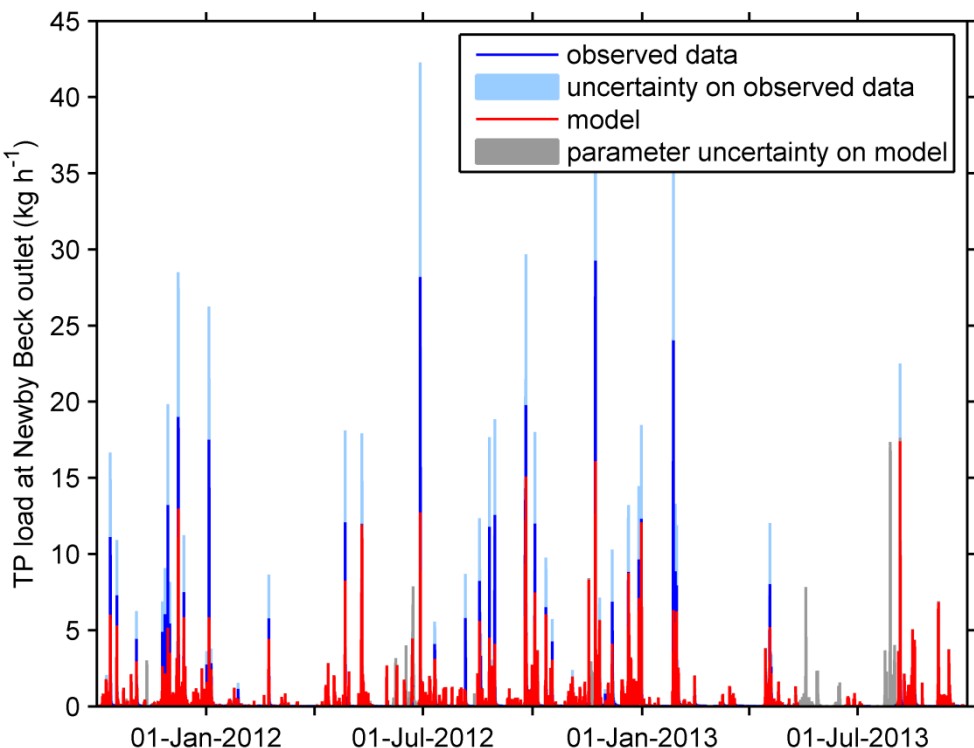

**Figure 5  First-order model between effective rainfall and total phosphorus (TP) load at Newby Beck for the identification period 1 October 2011 – 30 September 2013. Continuous-time model with structure [1, 1, 1] (see Table 3); $R_t^2 = 0.69$.  The light blue band indicates the 95% uncertainty bounds on the measurement data. The grey band indicates the 95% confidence limits on the parameter uncertainty (at this scale, only visible during periods where TP data are missing).  See Fig. 6 for zoomed in sections. Total model predictive uncertainty (including the residual uncertainty) is larger than parametric uncertainty and would enclose the observations most of the time.**

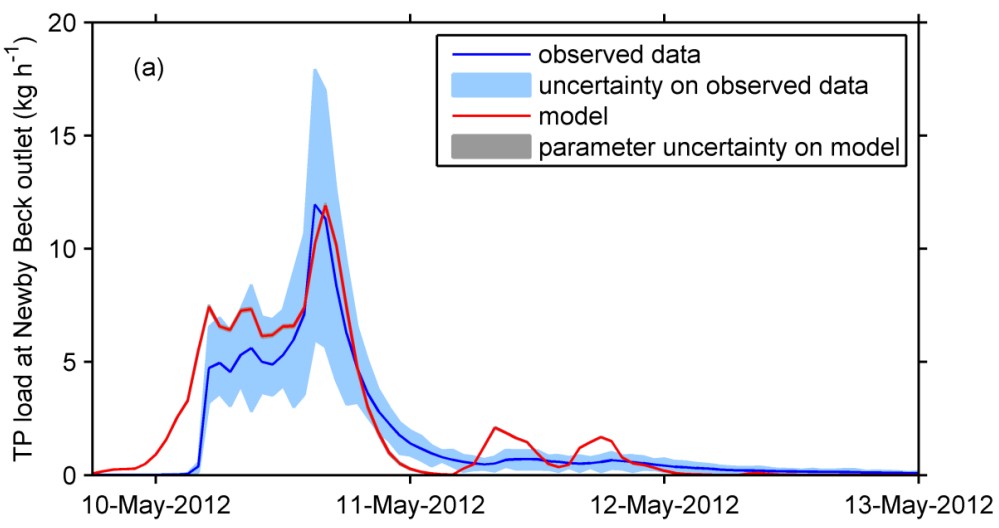

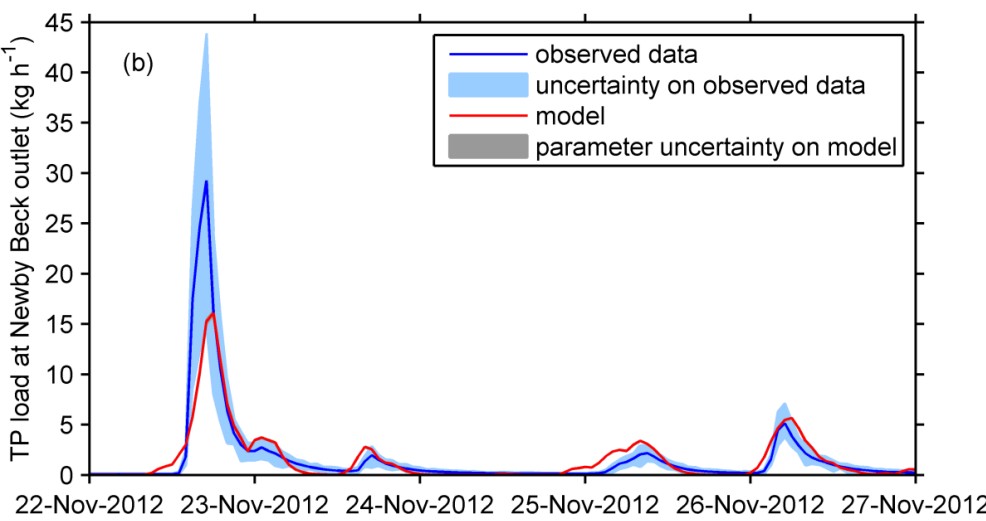

**Figure 6  First-order model between effective rainfall and total phosphorus (TP) load at Newby Beck, expanded from Fig. 5, for storm events in May 2012 (a) and November 2012 (b) . Continuous-time model with structure [1, 1, 1] (see Table 3); $R_t^2 = 0.69$. The light blue band indicates the 95% uncertainty bounds on the measurement data.  The grey band indicates the 95% confidence limits on the parameter uncertainty (at this scale, only visible during periods where TP data are missing).  Total model predictive uncertainty (including the residual uncertainty) is larger than parametric uncertainty and would enclose the observations most of the time.**

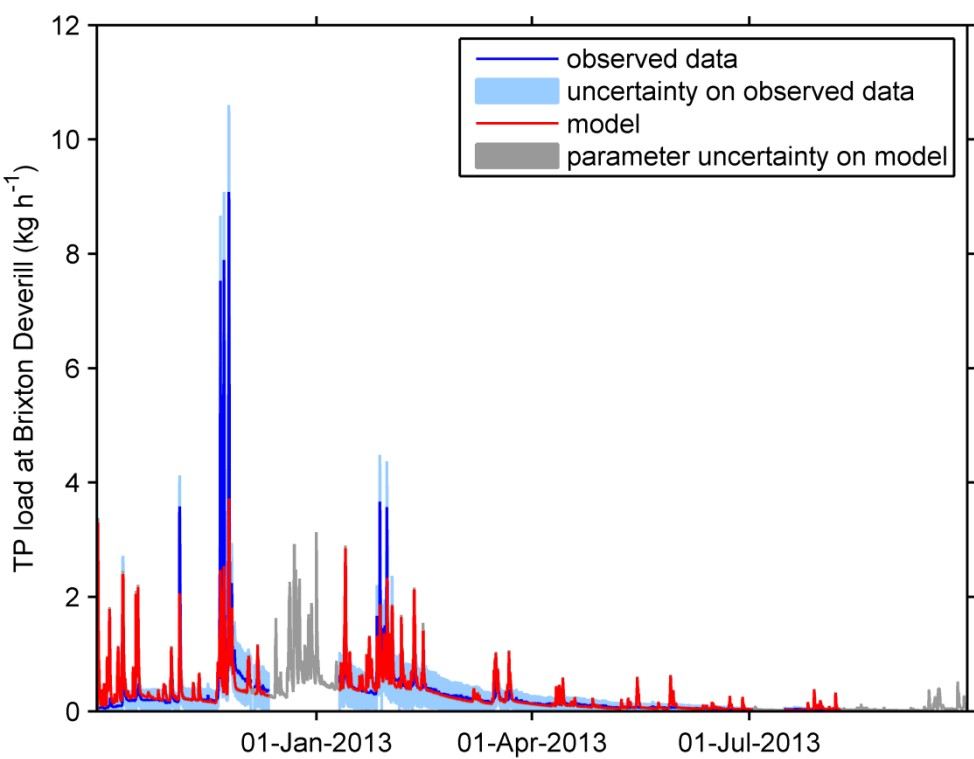

**Figure 7  Second-order model between effective rainfall and total phosphorus (TP) load at Wylye for the identification period 1 October 2012 – 30 September 2013.  Continuous-time model with structure [2, 2, 6] (see Table 3); $R_t^2 = 0.67$.  The light blue band indicates the 95% uncertainty bounds on the measurement data,  The grey band indicates the 95% confidence limits on the parameter uncertainty (at this scale, only visible during periods where TP data are missing).  Total model predictive uncertainty (including the residual uncertainty) is larger than parametric uncertainty and would enclose the observations most of the time. For zoomed in periods, see Fig. 8.**

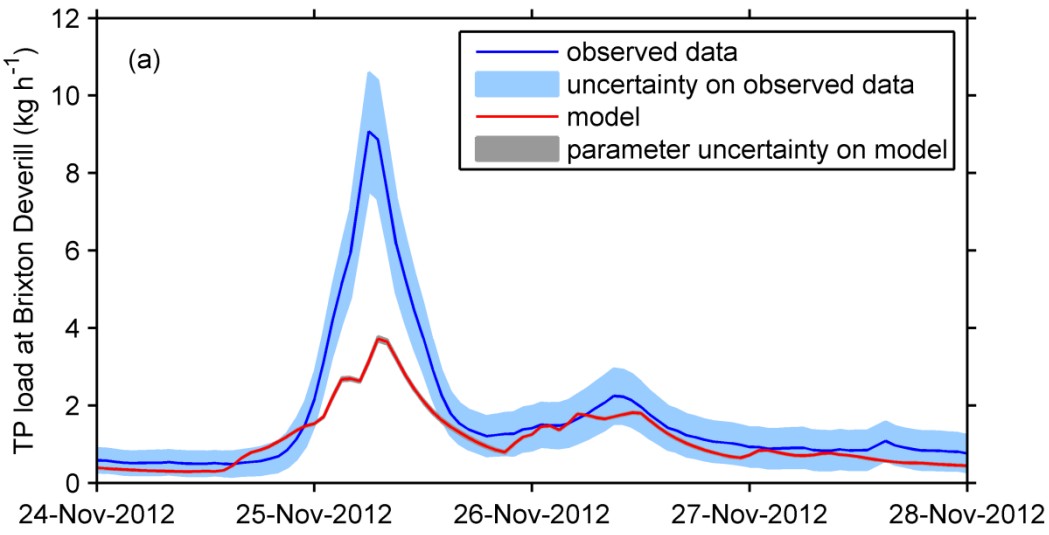

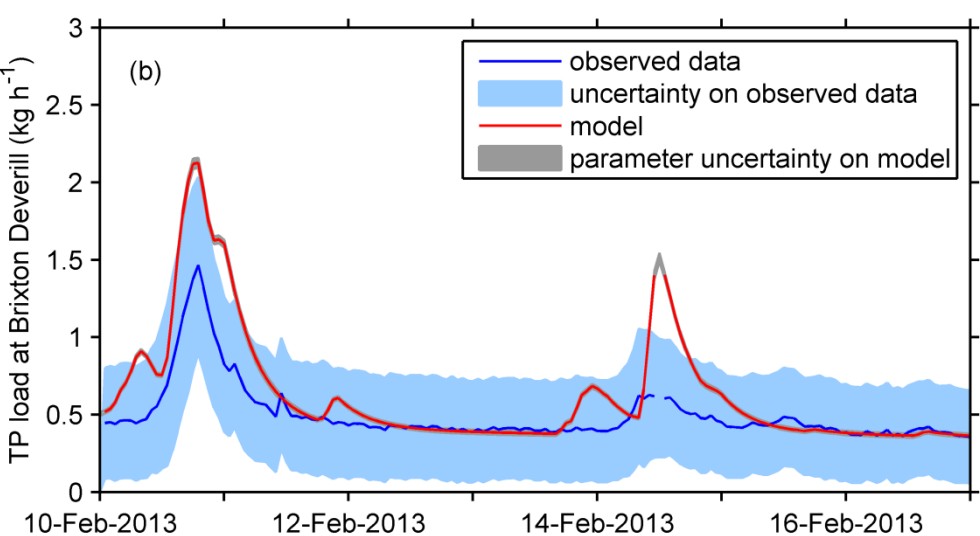

**Figure 8** **Second-order model between effective rainfall and total phosphorus (TP) load at Wylye for storm events in November 2012 (a) and February 2013 (b). Continuous-time model with structure [2, 2, 6] (see Table 3); $R_t^2 = 0.67$. The light blue band indicates the 95% uncertainty bounds on the measurement data, the grey band indicates the 95% confidence limits on the parameter uncertainty (at this scale, only visible during periods where TP data are missing). Total model predictive uncertainty (including the residual uncertainty) is larger than parametric uncertainty and would enclose the observations most of the time.**

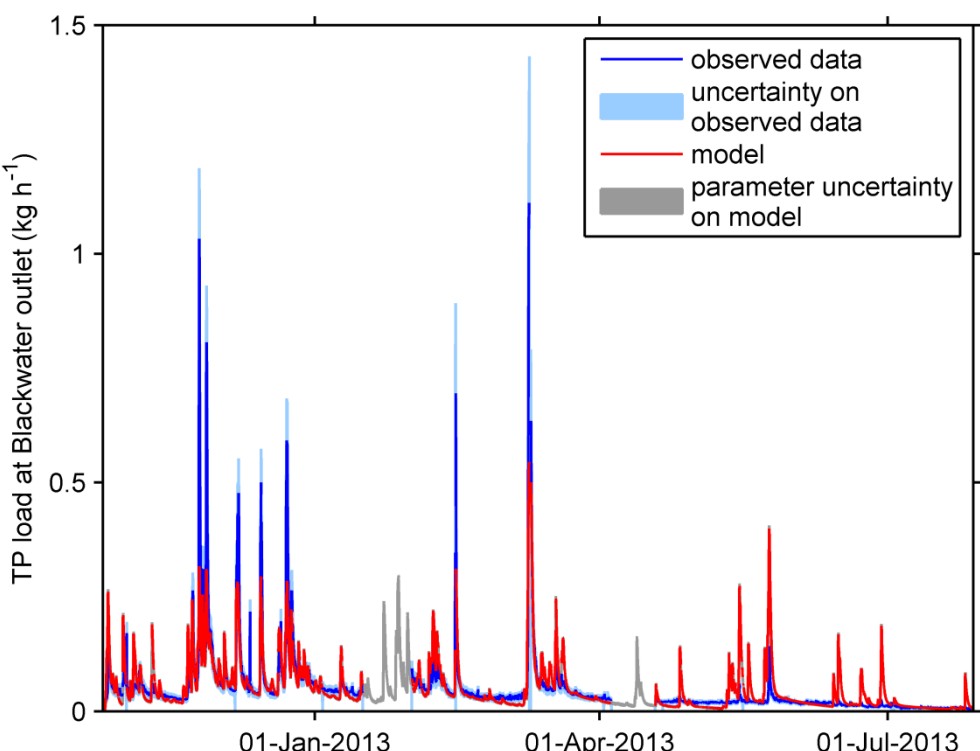

**Figure 9** **Second-order model between rainfall and total phosphorus (TP) load at Blackwater for the identification period 26 October 2012 – 28 July 2013. Continuous-time model with structure [2, 2, 4] (see Table 3); $R_t^2 = 0.67$. The light blue band indicates the 95% uncertainty bounds on the measurement data,  The grey band indicates the 95% confidence limits on the parameter uncertainty (at this scale, only visible during periods where TP data are missing).  Total model predictive uncertainty (including the residual uncertainty) is larger than parametric uncertainty and would enclose the observations most of the time. For zoomed in periods, see Fig. 10.**

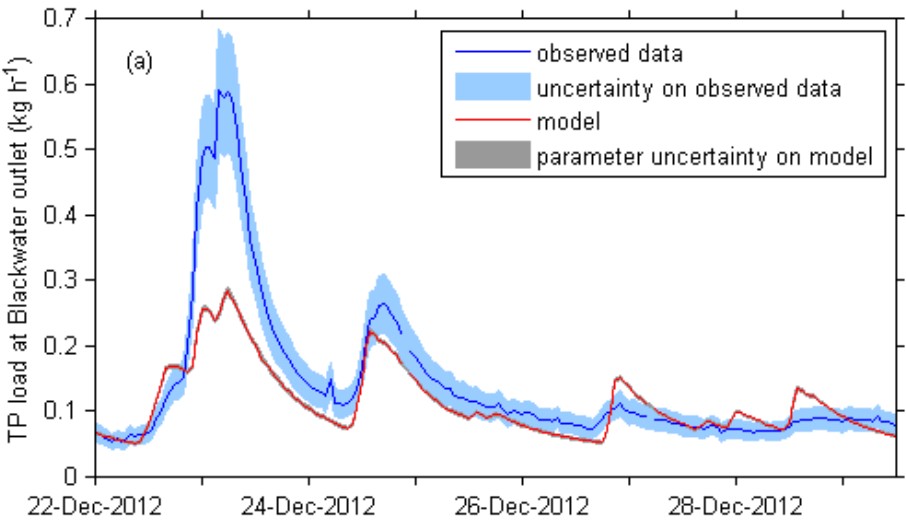

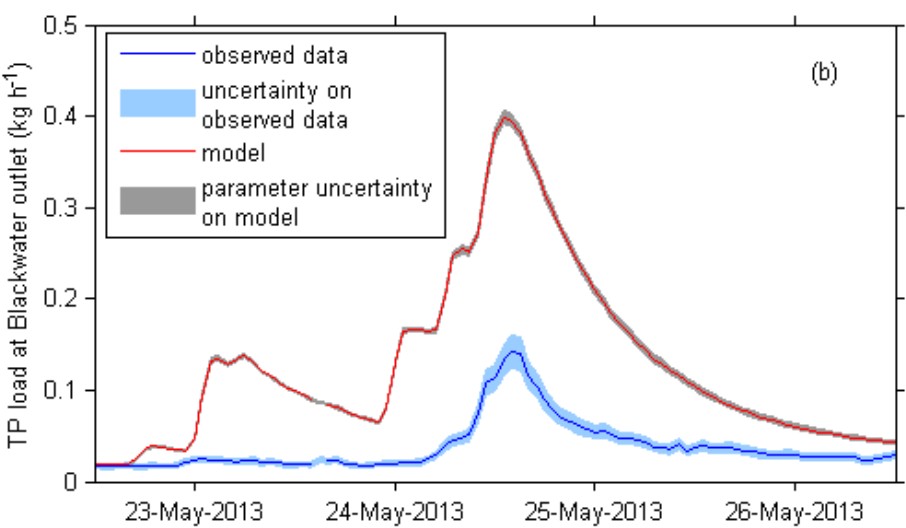

**Figure 10** Second-order model between rainfall and total phosphorus (TP) load at Blackwater for storms in December 2012 (a) and May 2013 (b). Continuous-time model with structure [2, 2, 4] (see Table 3); $R_t^2 = 0.67$. The light blue band indicates the 95% uncertainty bounds on the measurement data, The grey band indicates the 95% confidence limits on the parameter uncertainty (at this scale, only visible during periods where TP data are missing). Total model predictive uncertainty (including the residual uncertainty) is larger than parametric uncertainty and would enclose the observations most of the time.