# Peer review of "Prediction of storm transfers and annual loads with data-based mechanistic models using high-frequency data"

_Hydrology and Earth System Sciences, 2017_

## Referee Comment (RC1) · S. Stoll (Referee) · 13 Jun 2017

General remarks:

In the manuscript "Data-based mechanistic model of catchment phosphorus load improves prediction of storm transfer and annual loads in surface water" the authors present different data-based mechanistic models describing the dynamics of discharge and phosphorus load in three catchments in the UK.

Generally, I find the manuscript to be very interesting, well written and suitable for HESS (after some revisions). While I agree with the authors that DBM models are very
helpful in detecting dominant transfer modes I think that some of the alleged benefits of the modelling approach are overstated. For example, I doubt that these models can "help in planning appropriate pollution mitigation measures" as stated in the abstract. The reason for that is the nature of these models. The only input driving the models is rainfall (and sometimes discharge) data. Many features which are known to influence the phosphorus dynamics (like soil type, soil phosphorus concentration, management practices, tile drainage density, etc.) and which would be the primary entry point for any mitigation measures are not directly considered. Accordingly, the effect of any changes in these features (e.g. management practices) cannot be evaluated (not saying that physically-based models are per se any better with regard to that given the parameter uncertainty). In my opinion, the presented DBM models are much better suited to analyze the effects in changes of the precipitation (as rainfall is the main input) under the condition that these future precipitation conditions are covered in the calibration period.

In addition, I would love to see some more analysis of the very nice data they collected. I would assume that the manuscript would greatly benefit if the model results would be discussed together with the data (for example detailed analyses of the hysteresis curves).

Specific remarks:

Title: Improvement compared to what, other models?

P1, L31-32: See comments above

P2, L7: The authors correctly point out the importance of the measurement uncertainty. However, in the whole manuscript no information is provided regarding the uncertainty of the rainfall, discharge and phosphorus measurements or how this uncertainty is handled in the modelling approach. Especially the stage-discharge relationship (regarding the discharge measurements of flood events) can be subject to considerable uncertainty which would directly translate into uncertainty of the phosphorus loads. One

could argue that the measurement uncertainty is indirectly accounted for by the parameter uncertainty. However, given that the uncertainty bands are hardly detectable in the figures and measurements (without error bars) are not covered by it, it seems that either an important process is not captured by the model or that the measurement uncertainty is underestimated.

P2, L24-32: Here, the authors report the disadvantages and shortcomings of large, overparameterized process-based models (e.g. SWAT). I understand the motivation for that and even to a large degree agree with them. However, the authors should not only pick and describe the most extreme (or worst) process-based models. There are also parsimonious process-based models which can deliver reasonable results describing dynamics of phosphorus on hourly time steps (for example Hahn et al., 2013) or spatial herbicides losses (which have very similar transport pathways) (for example Frey et al., 2011) with few parameters.

P4, L14: What was the motivation to measure TP and not distinguish between or focus on particulate and/or dissolved phosphorus? Particulate (PP) and dissolved phosphorus (DP) can have different pathways and dynamics. While PP often shows a clockwise hysteresis (P peak before Q peak), DP often shows an anti-clockwise hysteresis (Q peak before P peak) (Dupas et al, 2015). By modelling them separately, it would be probably easier to identify a suitable transfer function and the corresponding pathways.

P5, L17: What is R in the equation, rainfall?

P6, L6-10: What is the motivation for setting up these short-term models for the Newby Beck catchment when the long-term model have similar performances and structures?

P6, L15: If I understand it correctly, the output which is used to identify and calibrate the model is also used as an input. I find this contra-intuitive and not really "proper". Why not use a precipitation based antecedent wetness index?

P7, L7-10: Some scatter plots would be very helpful to illustrate the Q-P relationships.

P7, L16: Table number is missing.

P7, L19-20: Because Blackwater has the lowest specific discharge. It would be good to discuss and explain the differences in the specific discharges and P concentrations among the catchments.

P7, L28-30: So were model results actually used to fill data gaps for the longterm model? If yes, this should be clearly stated and discussed accordingly.

P7, L29-30: How can model results help in identifying problems in the extrapolation of the stage-discharge relationships, when the whole model itself is based and calibrated with data of exactly these stage-discharge relationships? In my opinion the model is only reliable for the conditions covered during the calibration period. If more extreme events would be included in the calibration period, the model and the parameters would very likely be different.

P8, L4: Should be "Table 2"

P8, L4-18: I find the discussion and evaluation of storm Desmond a bit constructed and unnecessary. You don't need a DBM model to realize that discharge and P load was underestimated when there are reports of out-of-bank discharge bypassing the gauging station. The model also doesn't help in the quantification of the missed P and Q. As mentioned before, the model was trained under different conditions and is therefore in my opinion not really valid for very extreme cases not being part of the calibration period (again not saying that physically based models are any better).

Table 2: According to the time constants and order of the Q- and TP models, there are two pathways contributing to the discharge generation with only the fast pathway contributing to the TP generation. If I understand the concept of the TC correctly, TP reacts before the discharge rises. Is this in agreement with the measured data?

Table 3: What is the meaning of the term "using Qsim". If model outputs instead of actual measurements were used, this should be clearly stated, justified and discussed

(for example why is the performance worse for "using Qsim" that "using Qobs"?) In relation to that, how was TPLoad calculated? Did the authors used the modeled Q to calculate TPLoad or did they use the measured Q? Again, if modeled Q was used, this should be stated, justified and the consequences discussed.

Table 3 cont'd: For the Newby and Wylye TPLoad models effective rainfall was used as input, while regular rainfall was used for the discharge model. What is the meaning of that? Does it mean that for TP dynamics antecedent conditions are important, while they are not important for the discharge dynamics? Again, I would advise to discuss these findings as well as the different time constants and their percentages together with the actual measured data.

P9, L26: What does "effective rainfall (from the runoff model)" mean?

P11, L7: Same point again. What does "effective rainfall simulated by the rainfall-runoff model mean?

P12, L1: It's nice to have models with a low parameter uncertainty. However, when the uncertainty bands do not encompass the measurements, it's not really better situation than having a large parameter uncertainty. The model is either missing an important process or measurement uncertainty is not accounted for. A third reason could be a too narrow parameter sampling space in the MC method.

P12, L28-33: Although, the authors openly discuss the limitations of their modelling approach, there is one point I miss. They argue that understanding the rainfall-Q/TPLoad relationship through DBM models can help to identify dominant modes of the catchment and can therefore be used to target management interventions. I would argue that this is only possible if the identified dominant modes or pathways can be related to specific areas in the catchment. In my opinion it is not enough to know that 70% of the TPLoad was activated via a fast pathway. It is necessary to know which areas in the catchments are connected to the stream via this pathway, how these areas are managed and what their soil P status is. To actually plan and implement intervention

strategy, you need to know where (on which fields) and how to intervene. The "how" is strongly dependent on the "where". If you identified some fields with subsurface tile drainage as the contributing areas you would need a different intervention strategy as for example on a field with a tendency for surface runoff due to soil compaction. Knowing the temporal dynamics is not good enough, you would also need information about the spatial patterns.

Authorship:

I thought long about including this very last comment in the review. However, given the many discussions I had with colleagues about this very issue in the past, I feel somewhat obliged to mention that I find the number of authors contributing to this manuscript too excessive, given the nature of the article (a regular modelling study). I am very much in favor in acknowledging significant contributions (for example with respect to data gathering) with a co-authorship, however this seems not to be the case here. The authors state themselves that while two persons were responsible for the modelling, three persons did project management and the remaining fourteen (!) basically discussed the results and did some editing. I certainly don't want to offend any of the authors and obviously have no insights in the preparation process of the manuscript. Nonetheless, I would encourage each co-author to reflect if in their opinion they really contributed significantly to this manuscript.

References

Dupas, R., Gascuel-Odoux, C., Gilliet, N., Grimaldi, C., Gruau, G., 2015. Distinct export dynamics for dissolved and particulate phosphorus reveal independent transport mechanisms in an arable headwater catchment. Hydrol. Process. 29 (14), 3162e3178. http://dx.doi.org/10.1002/hyp.10432.

Frey,M.P.; Stamm,C.; Schneider,M.K.; Reichert,P. (2011) Using discharge data to reduce structural deficits in a hydrological model with a Bayesian inference approach and the implications for the prediction of critical source areas, Water Resources Research,

47(12), W12529 (18 pp).

Hahn,C.; Prasuhn,V.; Stamm,C.; Lazzarotto,P.; Evangelou,M.W.H.; Schulin,R. (2013) Prediction of dissolved reactive phosphorus losses from small agricultural catchments: Calibration and validation of a parsimonious model, Hydrology and Earth System Sciences, 17(10), 3679-3693.

––––––––––––––––––––––––––––––

---

## Referee Comment (RC2) · Anonymous Referee #2 · 30 Jun 2017

General Comments:

The manuscript "Data-based mechanistic model of catchment phosphorus load improves predictions of storm transfers and annual loads in surface waters" addresses relevant scientific questions and should be published after some major revisions. The highlight of the value of high temporal resolution measurements to capture the system dynamics is very valid, as well as the benefit of the chosen approach in determining the "optimal" temporal resolution for future measurement campaigns. However, especially the "methods" section needs to be expanded by clearly stating and justifying the assumptions made.

Specific comments:

Chapter 2.3:

What is exactly the meaning and the implications of not using a noise model in Eq 2? This should be explained in more detail. Any inference algorithm (in this case probably the RIV(C)BJ), needs to make assumptions about the errors to estimate parameters. Does not using a noise model mean that you assume the errors to be uncorrelated? Or is the error model inferred by the algorithm itself? The assumptions made in the inference process should be clearly stated and checked.

Eq 4 leads to significant violation of the mass balance w.r.t. water if Q(t-1) is larger than 1 (this depends strongly on the units of Q) and beta is larger than 0. This should be clearly stated, and then briefly mentioned why this is not a problem in this case (if it is not).

Chapter 3:

It would be interesting to validate some models with noise to get the total predictive uncertainty, not just parameter uncertainty. This would be a more meaningful validation and would allow actual statements about the uncertainty related to predictions of TP loads.

Most plots show observed and modeled quantities in the identification period. It would be interesting to have some more plots in the validation period (ev. including the total uncertainty bands including the noise). Also some zooms to specific time periods showing the strengths and the weaknesses of the models would be interesting.

Page 11 L33: The comparison of the quality of fit to process-based models needs a more detailed analysis. How exactly do you compare the quality of fit? With NS coefficients only? Is this justified, are the assumptions underlying NS coefficients fulfilled?

Technical corrections:

Page 8 L4: should be Table 2

Page 10 L27/28: one of the "fast pathway" should probably mean "slow pathway"?

Page 11 L22: "in" should be "an"

---

## Referee Comment (RC3) · Anonymous Referee #3 · 7 Jul 2017

General comments

This paper explores the use of Data-Based Mechanistic modelling to gain understanding of the dynamics of phosphorus transport using a high resolution TP concentration and load time series data set. To my knowledge, such a study has never been conducted using an hourly time series data set. As such, the paper will be useful to anyone working in phosphorus transport – a key component to understanding eutrophication issues. It is particularly worth noting that the authors are making the data openly available for which they are to be applauded. Presumably this will include the necessary meta-data to enable future users to understand the uncertainty in the measurements.
Overall, the paper is very well written, and suitable for publication after the issues noted below have been dealt with.

Specific comments

Given that approximately 3 years of data have been used in this study, could the authors comment on the magnitude of the error/uncertainty in the result if only daily data were used, or even worse, data collected at varying intervals (as is frequently the case)?

The authors cite Young (2010) who argued that the sampling period should be at most 1/6th of the quickest time constant. Given this and finding that the time constant for the fast pathway varied between 2 and 15 hours, using hourly data at the low end of this scale would mean the sampling frequency would be too slow – questioning the statement in the abstract that hourly data are "necessary". For time constants near 2 hours, hourly data would be inadequate by the condition given by Young (2010) – does this mean that the authors are revising the recommendation given by Young (2010)? My thought is this is not the case, so the phrasing in the abstract seems a little strange.

Page 2, line 25: The authors state that USLE is a process-based model. In a broad sense it could be called this, but I feel it would be more accurate to describe it as an empirically-based model.

Page 4, line 23: Can visual inspection really determine whether there has been a significant loss of information? Was the inspection merely looking at the hydrograph (i.e. plotting the data)? Could some objective measure be developed for this?

Page 6, line 26: A minor point, but there are some issues with the description of NSE. The numerator is the variance of the model residuals if and only if the mean residual (i.e. bias) is identically zero. It is a reasonable approximation to the variance if the bias is small, however it is a biased estimate as it will always be a little larger than the actual variance. On page 8 (line 22), the authors state that the model bias was less than $\pm$ 10% for all three catchments. What would the impact of this amount of model

bias be on the comparison between $R_t^2$ and NSE? An even more minor point is that the normalisation factor for the variance is usually 1/(N-1) rather than (1/N).

Page 7, line 25: Text seems to jump a little at this point. Maybe add a sentence bringing "storage" into the picture before the closing sentence for this paragraph?

Page 8, line 11: Presumably, the rating curve would need to be extrapolated to get to this flow level, which would also contribute to the uncertainty in the estimated discharge?

Technical corrections

Page 7, line 14: space missing after Avon

Page 9, line 23+24: The flow fractions for the slow flow component are not really needed as this are just 100

Page 9, line 32: Space missing after "drain flow"

Page 12, line 7: space missing after "have"

Page 12, line 31: space missing before "Yang"

---

## Referee Comment (RC4) · Anonymous Referee #4 · 8 Jul 2017

**General comments**

The manuscript presents data-based models (DBMs) of discharge (Q) and total phosphorus (TP), with rainfall as the driver. For short-term description, rainfall (R) was the input, while for longer periods an effective rainfall intensity accounted for possible nonstationary behaviour. Effective rainfall was derived from R using a modifying factor that was a power function of the most recent discharge. The models were applied to 3 small agricultural catchments in the UK, where an exceptional monitoring programme provided high-resolution time-series of R, Q, and TP. For short term, model calibration was successful only for the most flashy catchment, while slow trends and data gaps

prevented calibration for the other two catchments. Long-term calibration was satis-factory in each catchment. Validation was unsuccessful for long-term TP modeling in some cases because the effective rainfall relationship failed to properly adjust R data beyond the training phase. Model calibration/identification yielded the time-constants and relative weight of the 'fast' and 'slow' response components. The paper is well written and shows an interesting analysis of an extraordinary dataset with sophisti-cated modeling tools. Besides these merits, the manuscript would clearly benefit from a new title, a somewhat more balanced judgement on data-based models, a clearer model introduction and a better separation from Ockenden et al. (in press).

The title promises improving "predictions of storm transfers and annual loads in surface waters", yet the outcome was rather "a better understanding of the dominant nutrient transfer modes, which will, in-turn, help in planning appropriate pollution mitigation measures" (last sentence of abstract). I agree more, but not completely, with the ab-stract: the manuscript is about transfer modes derived from observed data and not better predictions (the predictive power of the models was rather limited). Loads did not get too much emphasis either. Therefore, I suggest to change the title to match the main findings.

According to the authors, a major advantage of the data-based approach is that rather complex models can be used to predict Q and TP without the need to know anything about the major processes – knowledge will be extracted from the data. Indeed, com-plexity of their model rivals that of certain conceptual models. However, this application wasn't a clear success. The intro and the conclusions are very optimistic about data-based models. However, such DBMs have the same problems as conceptual models: their elements are so abstract that they can't be linked to anything observable. This semi-black-box behaviour is nicely demonstrated in the manuscript: a strong 'slow' component is present in TP in certain catchments, various possible reasons are listed, but there is no way to know which applies in the specific case (e.g. SRP in base-flow may indeed come from WWTPs or activated deposits, etc, but which case does

apply here? [We are not informed whether there are WWTPs in the catchments.]).
So while process-based models are typically overparameterised and laden with un-
certainty, their less abstract formulation leaves open at least theoretically to gather
additional observations to prove or falsify hypotheses. Overall, I think that it should be
mentioned that DBMs pay with an extreme data demand for not asking a priori knowl-
edge on the system. Given that unresolvable issues appeared with such an extremely
good data coverage, it is somewhat dissonant to recommend DBMs for catchment
management when (i) model constituents can't be linked to anything else than the in-
put, and (ii) there are practically no other catchments in the World with a comparable
data coverage.

Another issue is related to the model presentation and the relation to the paper by
Ockenden et al. (in press). The Methods section provides a very brief overview of the
models referring to the companion paper for details and for calibration, so the models
were first published and calibrated therein (this can't be verified because the article
isn't accessible at the moment, it's still in press). Please highlight the novel parts of this
manuscript compared to Ockenden et al. (in press). If the main novelty is the dynamics
of TP load, the analysis of the results could be somewhat extended at the cost of
details on the fast/slow components of TP (which are presented to the last detail). For
example, it would be useful to elaborate more on the TPflux vs Q relationship. Yes, Q
was already used to calculate TPflux, but the final correlation is actually determined
by the relative variance of Q and TPconc. This would highlight how much delay and
nonlinearity (both causing hysteresis) is present and therefore how much we gain by
having a nonlinear autoregressive model.

Independent of this, deriving the applied models from the general 2nd order continu-
ous transfer function (TF) model seems to be an unnecessarily complicated choice for
several reasons:

- Only those models were accepted, which could be converted to the parallel linear

storage format. This is very welcome, because such models are Markovian, i.e. the system's current internal state (or the last state in discrete formulation) and the current inputs completely determine the system's response. This assumption is typically made in most environmental and hydrological models. In contrast, a 2nd order TF model can be non-Markovian too (=a long system history is required to understand the current response, actual state is not a complete descriptor), which would be very hard to justify (which physical/biological/chemical process would lead to such system?).

• Continuity dominates in the model description, while eq. 4 and the aggregation to 30 minutes make it obvious that inputs were treated discretely. Then why bother with the more complex continuous models? These kinds of models are not used very frequently in hydrology/water quality modeling (as opposed to signal processing). Potential readers may easier understand if 2 parallel linear storages or ARX models were mentioned as alternative formulations for the same model.

• If the parallel storage formulation is so important to learn about slow and fast components, why are parameters shown in the general 2nd order form in Table S5?

**Smaller issues**

Page 2 Line 25: USLE is more semi-empirical than process-based.

Page 4 Equation 1: Use consistent units. If Q had [m3/h], and TPconc [kg/m3], TPload would readily be in [kg/h] without a conversion constant.

Page 5: Were the tau (delta) delay constants calibrated or fixed?

Page 6 L 15: If Re was necessary because the internal state of the catchment affected runoff and TP transport, would it make sense to use Re to the model Q as well?

Page 7 L 22-25: Of course, most pollutants do not follow Q, because they have either

limited or temporarily activated sources or they partition between water and sediment. According to your argument on celerity, even non-partitioning conservative pollutants would theoretically show a hysteresis.

Page 9 L 3-4: Converting these constants to half-lives would make them easier to judge. It is somewhat difficult to grasp decay to 1/exp(1).

Page 10 L 18-21: If we don't know the mechanisms responsible for the slow pathway, what kind of measures could be taken?

Page 12 L 24-28: It's true that process-based models make some assumptions that do not always hold, but here it was demonstrated that neither the DBM can always be validated. Time-variable parameters are a useful concept, but seldom implemented.

As fluxes of TP are modelled, Fig S1 should rather show TP fluxes against Q. This could reduce clutter and illustrate how a naive linear model would work. Considering my comment above, it would be useful to move a modified version of this figure into the main text.

A major model-based finding of this study is the demonstration of are the importances of fast and slow pathways of Q and TP in the catchments. This could, at least partially, be derived directly from the data! High baseflow indices and slower recession indicate important slow pathways of Q, high baseline concentration indicates the same for TP. As the applied model doesn't have any mechanistic explanatory power (e.g. identification of reasons for these), how could management benefit from modeling? Please comment on this briefly.

On load figures, load is [kg], but per which time unit?

What are the time units in e.g. Table 2? If the continuous version was used, time has to have a unit. If the discrete version was used, the applied timestep has to be written.

What are the other units in Table 2?

---

## Author Comment (AC1) · 25 Aug 2017

Authors' response to Reviewer 4, Anonymous
For clarity, we have included the reviewer's comments in black; our response is in blue

**General comments**
The paper is well written and shows an interesting analysis of an extraordinary dataset with sophisticated modeling tools.
Thank you

Besides these merits, the manuscript would clearly benefit from a new title, a somewhat more balanced judgement on data-based models, a clearer model introduction and a better separation from Ockenden et al. (in press).
See later comment on Ockenden et al. (now published)

The title promises improving "predictions of storm transfers and annual loads in surface waters", yet the outcome was rather "a better understanding of the dominant nutrient transfer modes, which will, in-turn, help in planning appropriate pollution mitigation measures" (last sentence of abstract).
We accept the comments on the comparison with other models and we propose to revise the title to "Prediction of storm transfers and annual loads with data-based mechanistic models using high-frequency data".

I agree more, but not completely, with the abstract: the manuscript is about transfer modes derived from observed data and not better predictions (the predictive power of the models was rather limited). Loads did not get too much emphasis either. Therefore, I suggest to change the title to match the main findings.
We accept the comments on the limitations of the model, but we maintain that the models have very good predictive power where there have been no fundamental changes to the catchment, and under similar input conditions. We propose to change the title as above. We propose to tone down the abstract and text to say "The models led to a better understanding of the dominant transfer modes, which will be helpful in determining phosphorus transfers following changes in precipitation patterns in the future."

According to the authors, a major advantage of the data-based approach is that rather complex models can be used to predict Q and TP without the need to know anything about the major processes – knowledge will be extracted from the data.
The advantage of the data-based approach is that rather simple models can be used. The advantages and limitations of the DBM modelling approach are listed in Table 4.

Indeed, complexity of their model rivals that of certain conceptual models. However, this application wasn't a clear success. The intro and the conclusions are very optimistic about databased models. However, such DBMs have the same problems as conceptual models: their elements are so abstract that they can't be linked to anything observable. This semi-black-box behaviour is nicely demonstrated in the manuscript: a strong 'slow' component is present in TP in certain catchments, various possible reasons are listed, but there is no way to know which applies in the specific case (e.g. SRP in baseflow may indeed come from WWTPs or activated deposits, etc, but which case does apply here? [We are not informed whether there are WWTPs in the catchments.]). So while process-based models are typically overparameterised and laden with uncertainty, their less abstract formulation leaves open at least theoretically to gather additional observations to prove or falsify hypotheses.
We accept that there are limitations to what you can interpret from DBM models – such as the spatial differences within a catchment. But we maintain that these issues (such as the WWTP

sources mentioned above) are exactly the same as with process-models, except that for those models you have to hypothesise first in order to decide what processes to include in the model. There is also a danger in process-based models that one could draw conclusions about the components of the system dynamics not present in the data (and so not indentifiable), thus using model artefacts (or noise patterns within uncertainty) to draw such conclusions.  This assists in getting the right answers for the right reasons (Kirchner, 2006).
Kirchner, J. W.: Getting the right answers for the right reasons: Linking measurements, analyses, and models to advance the science of hydrology, 42, W03S04, doi:10.1029/2005wr004362, 2006.

Overall, I think that it should be mentioned that DBMs pay with an extreme data demand for not asking a priori knowledge on the system. Given that unresolvable issues appeared with such an extremely good data coverage, it is somewhat dissonant to recommend DBMs for catchment management when (i) model constituents can't be linked to anything else than the input, and (ii) there are practically no other catchments in the World with a comparable data coverage.
The high data demand of DBM models is noted in Table 4 and the section on advantages/limitations of the modelling method.   The DBM transfer function (TF) models link the observations of inputs and outputs in a causal structure where the model is a system of differential equations (normally, but not necessarily linear) picking up the dominant dynamic modes present in the data. This is not a black box model, as the model structure is selected in the modelling process as a physically feasible one (hence the Mechanistic in the name) and identified from the data, which eliminates the modes of behaviour that are not present in the data, and thus not identifiable (instead contributing to the uncertainty). The basic component of TF models – the well known ADZ model for flow-flow models but also for rainfall-runoff models (Young, Beven, others) is based on mass balance between the input and output signals (flow, rainfall), the structure of such mass transfer blocks connection (parallel, serial) is also based on physical considerations, just based on other equations/paradigms compared to physics/process based models.
Just because data coverage in many catchments is poor does not mean it should always stay that way. Technology and monitoring methods are improving all the time so that high-frequency data are now more readily available, e.g. Jordan et al., 2005, 2007; Outram et al., 2014; Skeffington et al., 2015.  We should embrace efforts to improve data coverage and ways to use it wisely.
Jordan, P., Arnscheidt, J., McGrogan, H., and McCormick, S.: High-resolution phosphorus transfers at the catchment scale: the hidden importance of non-storm transfers, Hydrol. Earth Syst. Sci., 9, 685-691, 2005.
Jordan, P., Arnscheidt, A., McGrogan, H., and McCormick, S.: Characterising phosphorus transfers in rural catchments using a continuous bank-side analyser, Hydrol. Earth Syst. Sci., 11, 372-381, 2007.
Outram, F. N., Lloyd, C. E. M., Jonczyk, J., Benskin, C. M. H., Grant, F., Perks, M. T., Deasy, C., Burke, S. P., Collins, A. L., Freer, J., Haygarth, P. M., Hiscock, K. M., Johnes, P. J., and Lovett, A. L.: High-frequency monitoring of nitrogen and phosphorus response in three rural catchments to the end of the 2011-2012 drought in England, Hydrol. Earth Syst. Sci., 18, 3429-3448, 10.5194/hess-18-3429-2014, 2014.
Skeffington, R. A., Halliday, S. J., Wade, A. J., Bowes, M. J., and Loewenthal, M.: Using high-frequency water quality data to assess sampling strategies for the EU Water Framework Directive, 19, 2491-2504, 10.5194/hess-19-2491-2015, 2015.

Another issue is related to the model presentation and the relation to the paper by Ockenden et al. (in press). The Methods section provides a very brief overview of the models referring to the companion paper for details and for calibration, so the models were first published and calibrated therein (this can't be verified because the article isn't accessible at the moment, it's still in press). Please highlight the novel parts of this manuscript compared to Ockenden et al. (in press).

**This paper is a companion paper to Ockenden et al. (Nature Comms, 2017), now published. That paper uses a DBM model as part of a multi-model study to predict phosphorus transfers in the future, i.e. an application of the model developed in this paper. This paper provides full details of calibration and validation and, particularly, interpretation of results, which is not included in Ockenden et al., 2017. However, we propose to expand the methods section here to include full details of the models.**

If the main novelty is the dynamics of TP load, the analysis of the results could be somewhat extended at the cost of details on the fast/slow components of TP (which are presented to the last detail). For example, it would be useful to elaborate more on the TPflux vs Q relationship. Yes, Q was already used to calculate TPflux, but the final correlation is actually determined by the relative variance of Q and TPconc. This would highlight how much delay and nonlinearity (both causing hysteresis) is present and therefore how much we gain by having a nonlinear autoregressive model. **There two types of lags: pure time delay (the 'Poohstick time' for the flow-flow models) and dynamic lag resulting from mass transfer dynamics, both contributing to the observed hysteresis loop but resulting from different phenomena, not easily distinguishable from the shape of hysteresis, but perfectly identifiable and quantifiable in the DBM/TF approach. Relative variance? That would indicate a static relationship between Q and TPconc. What we established from the data is that this relationship is dynamic.**

Independent of this, deriving the applied models from the general 2nd order continuous transfer function (TF) model seems to be an unnecessarily complicated choice for several reasons:
• Only those models were accepted, which could be converted to the parallel linear storage format **(or serial, or indeed first order depending on the structure identification results).**
This is very welcome, because such models are Markovian, i.e. the system's current internal state (or the last state in discrete formulation) and the current inputs completely determine the system's response. This assumption is typically made in most environmental and hydrological models. In contrast, a 2nd order TF model can be non-Markovian too (=a long system history is required to understand the current response, actual state is not a complete descriptor), which would be very hard to justify
**Any order TF model has a range of equivalent state space Markov type models, which is clearer in their discrete form, but the discrete and continuous forms of TF models are equivalent. We are not sure what the Reviewer is referring to here.**

(which physical/biological/chemical process would lead to such system? – **any uncertain mixing or transport process).**

• Continuity dominates in the model description, while eq. 4 and the aggregation to 30 minutes make it obvious that inputs were treated discretely
**Continuous time models are estimated from sampled data – there is no contradiction here.**
Then why bother with the more complex continuous models?
**Continuous time models are more numerically robust and have a direct interpretation as systems of differential equations (Young, 2011)**
**Young, P. C.: Recursive Estimation and Time-Series Analysis: An Introduction for the student and practitioner, Second ed., Springer, New York, 504 pp., 2011.**

These kinds of models are not used very frequently in hydrology/water quality modeling (as opposed to signal processing)

**They have not been used widely until recently because of the lack of effective model identification methods such as those in the CAPTAIN Toolbox – this is a part of the novelty of this work.**

Potential readers may easier understand if 2 parallel linear storages
or ARX models were mentioned as alternative formulations for the same model.
**This is the case for discrete time models, which have no direct differential equation interpretation and are less robust with respect to stiff systems with dynamic modes of very different dynamics (Young, 2011).**
**Young, P. C.: Recursive Estimation and Time-Series Analysis: An Introduction for the student and practitioner, Second ed., Springer, New York, 504 pp., 2011.**

• If the parallel storage formulation is so important to learn about slow and fast
components, why are parameters shown in the general 2nd order form in Table S5?
**Table S5 shows the general polynomial form for information (and in case anyone wants to simulate using this model), because they are estimated in this form, so their parameter uncertainties are obtained in the general polynomial form too. Factorisation of the rational polynomial TF into parallel, serial etc. components is the next step in the DBM process. The decomposed form, with time constants for slow and fast components are given in Table 3.**

**Smaller issues**
Page 2 Line 25: USLE is more semi-empirical than process-based.
**Accepted. This will be changed**

Page 4 Equation 1: Use consistent units. If Q had [m3/h], and TPconc [kg/m3], TPload
would readily be in [kg/h] without a conversion constant.
**This is true, but would then result in working with values that are either very large (in the case of Q) or very small (in the case of TPconc). We felt that it was better to stick to the units in which the variables were measured.**

Page 5: Were the tau (delta) delay constants calibrated or fixed?
**Estimated from data using information criteria, just as for the model structures.**

Page 6 L 15: If Re was necessary because the internal state of the catchment affected
runoff and TP transport, would it make sense to use Re to the model Q as well?
**Re was used in all models except the Blackwater rainfall-TPload model. However, we accept that this does not come across clearly in the text. Q is modelled as a linear transfer function with Re as input, where the non-linear relationship between R and Re is estimated at the same time as the TF parameter estimation. TPload is also modelled as a linear transfer function with Re as input, except that Re (and Q) in this case are first simulated using the parameters previously estimated for the R-Q model.**

Page 7 L 22-25: Of course, most pollutants do not follow Q, because they have either limited or temporarily activated sources or they partition between water and sediment. According to your argument on celerity, even non-partitioning conservative pollutants would theoretically show a hysteresis.
**A truly 'non-partitioning conservative pollutant' would act as a conservative tracer, moving exactly with the water particles (which forms the basis for dilution gauging using conservative tracers). In this case there would be no hysteresis on the phase-plot between Q and that hypothetical tracer. This does not contradict our comment on the hydrograph representing the integrated effects of celerities.**

Page 9 L 3-4: Converting these constants to half-lives would make them easier to judge. It is somewhat difficult to grasp decay to 1/exp(1).

**The half-life interpretation is only a good illustration for pure recession curves (with the ln(2) proportion between $T_c$ and $T_{1/2}$), it is less obvious for differential equations with changing complicated inputs. This is the standard definition of a time constant in a first order linear time-invariant dynamic process e.g. $A(t) = A_0 \exp(-t/T_c)$ where $T_c$ is the time constant, commonly used in hydrological literature (see Nash cascade definition, Shaw et al, 1994 etc).**

**Shaw, E. M.: Hydrology in Practice (3rd edition), Chapman and Hall, London and New York, 1994.**

Page 10 L 18-21: If we don't know the mechanisms responsible for the slow pathway, what kind of measures could be taken?

**These models do not provide recipes or final answers, but objectively point out specific parts of the system dynamics. Having identified that a slow pathway is so important, measures which prevent pollutants getting to the slow pathway in the first place, such as reductions at source, will be helpful. This may require further specific measurements, such as testing P in soils or identifying septic tanks in the catchment. The difference between DBM and process based models is that this interpretation in DBM models is made a posteriori, after the data assimilation and is based on objectively identified quantitative features of the process, with process based models the interpretation is done a priori, with all the caveats related to such a sequence.**

Page 12 L 24-28: It's true that process-based models make some assumptions that do not always hold, but here it was demonstrated that neither the DBM can always be validated. Time-variable parameters are a useful concept, but seldom implemented. As fluxes of TP are modelled, Fig S1 should rather show TP fluxes against Q. This could reduce clutter and illustrate how a naive linear model would work. Considering my comment above, it would be useful to move a modified version of this figure into the main text.

**The plots of Q against concentration are shown to illustrate the hysteresis loops and to show the background concentration at varying baseflows. However, we propose to add plots of Q against TPload to the supplementary information but do not feel that this would add to the main text. A linear dynamic model does not have to be naïve if it is identified from the data using a rigorous procedure. It indicates that this is the maximum model approximating the data well, that can be estimated based on the present data set.**

A major model-based finding of this study is the demonstration of are the importances of fast and slow pathways of Q and TP in the catchments. This could, at least partially, be derived directly from the data! High baseflow indices and slower recession indicate important slow pathways of Q, high baseline concentration indicates the same for TP.

**There are many, often very arbitrary methods of base flow estimation and there is much discussion in the literature as to which one is better, such heuristic dominance 'approximation' has neither rigorous quantification nor uncertainty estimation elements, unlike the DBM modelling procedures.**

As the applied model doesn't have any mechanistic explanatory power (e.g. identification of reasons for these), how could management benefit from modeling? Please comment on this briefly.

**See response as for Page 10 L 18-21 query.**

On load figures, load is [kg], but per which time unit?

**All units are hours. Load figures will be changed to kg h$^{-1}$.**

What are the time units in e.g. Table 2? If the continuous version was used, time has

to have a unit. If the discrete version was used, the applied timestep has to be written.
**Time units are hours.  This will be added to Table 2 (and Table 3).**

What are the other units in Table 2?
**Σobs and Σmodel are totals for the period of Storm Desmond.  These are in mm for runoff and kg for TPload.  These will be added to Table 2.**

---

## Author Comment (AC2) · 25 Aug 2017

Authors' response to Reviewer 3, Anonymous
For clarity, we have included the reviewer's comments in black; our response is in blue

General comments
This paper explores the use of Data-Based Mechanistic modelling to gain understanding of the dynamics of phosphorus transport using a high resolution TP concentration and load time series data set. To my knowledge, such a study has never been conducted using an hourly time series data set. As such, the paper will be useful to anyone working in phosphorus transport – a key component to understanding eutrophication issues. It is particularly worth noting that the authors are making the data openly available for which they are to be applauded. Presumably this will include the necessary meta-data to enable future users to understand the uncertainty in the measurements. Overall, the paper is very well written, and suitable for publication after the issues noted below have been dealt with.
**We are very grateful for this recognition of our contribution**

Specific comments
Given that approximately 3 years of data have been used in this study, could the authors comment on the magnitude of the error/uncertainty in the result if only daily data were used, or even worse, data collected at varying intervals (as is frequently the case)?
**The errors resulting from sampling well below the catchment dynamics have been well documented elsewhere, e.g. Lloyd et al., 2016; Johnes, 2007; Jones et al., 2012, Moatar et al., 2013. However, we propose to include these references.**
**Lloyd, C. E. M., Freer, J. E., Johnes, P. J., Coxon, G., and Collins, A. L.: Discharge and nutrient uncertainty: implications for nutrient flux estimation in small streams, Hydrol. Process., 30, 135-152, 10.1002/hyp.10574, 2016.**
**Johnes, P. J.: Uncertainties in annual riverine phosphorus load estimation: Impact of load estimation methodology, sampling frequency, baseflow index and catchment population density, J. Hydrol., 332, 241-258, 10.1016/j.jhydrol.2006.07.006, 2007.**
**Jones, A. S., Horsburgh, J. S., Mesner, N. O., Ryel, R. J., and Stevens, D. K.: Influence of Sampling Frequency on Estimation of Annual Total Phosphorus and Total Suspended Solids Loads, 48, 1258-1275, 10.1111/j.1752-1688.2012.00684.x, 2012.**
**Moatar, F., Meybeck, M., Raymond, S., Birgand, F., and Curie, F.: River flux uncertainties predicted by hydrological variability and riverine material behaviour, 27, 3535-3546, 10.1002/hyp.9464, 2013.**
The authors cite Young (2010) who argued that the sampling period should be at most 1/6th of the quickest time constant. Given this and finding that the time constant for the fast pathway varied between 2 and 15 hours, using hourly data at the low end of this scale would mean the sampling frequency would be too slow – questioning the statement in the abstract that hourly data are "necessary". For time constants near 2 hours, hourly data would be inadequate by the condition given by Young (2010) – does this mean that the authors are revising the recommendation given by Young (2010)? My thought is this is not the case, so the phrasing in the abstract seems a little strange.
**The reviewer is correct in that we are not revising Young's recommendation. Wherever possible we used continuous time models, which are less critical in this respect. However, we propose to edit the statement in the abstract to tone down the requirement for hourly data and state "high temporal resolution data are necessary to capture the dynamic responses in small catchments".**
Page 2, line 25: The authors state that USLE is a process-based model. In a broad sense it could be called this, but I feel it would be more accurate to describe it as an empirically-based model.
**Accepted. We will change this to semi-empirical**

Page 4, line 23: Can visual inspection really determine whether there has been a significant loss of information? Was the inspection merely looking at the hydrograph (i.e. plotting the data)? Could some objective measure be developed for this?

**There is no objective measure for this. Close visual inspection/comparison is very valuable in identifying whether the dynamics are still captured by the lower resolution data**

Page 6, line 26: A minor point, but there are some issues with the description of NSE. The numerator is the variance of the model residuals if and only if the mean residual (i.e. bias) is identically zero.

**Accepted. The transfer function (TF) estimates using Instrumental Variable methods produce unbiased parameter estimates; the bias of residuals is only asymptotically zero. NS measure with all its shortcomings is still a commonly used model metric. However, we propose to replace the word 'variance' with 'variance estimate' in the descriptions of NSE.**

It is a reasonable approximation to the variance if the bias is small, however it is a biased estimate as it will always be a little larger than the actual variance. On page 8 (line 22), the authors state that the model bias was less than ± 10% for all three catchments. What would the impact of this amount of model bias be on the comparison between R2t and NSE?

**The difference between $R_t^2$ and NSE is that NSE can be equivalent to either $R_t^2$ or $R^2$ depending on the application.**

An even more minor point is that the normalisation factor for the variance is usually $1/(N-1)$ rather than $(1/N)$.

**This is indeed a minor issue given the number of samples (thousands). However, we propose to replace the word 'variance' with 'variance estimate' in the descriptions of NSE and Rt2 to address this query.**

Page 7, line 25: Text seems to jump a little at this point. Maybe add a sentence bringing "storage" into the picture before the closing sentence for this paragraph?

**Accepted. We propose to add (before the last sentence of paragraph) "This indicates that antecedent conditions and the storage state of the catchment are important in determining the response."**

Page 8, line 11: Presumably, the rating curve would need to be extrapolated to get to this flow level, which would also contribute to the uncertainty in the estimated discharge?

**Exactly. A review of measurement data uncertainty is presented by McMillan et al., 2012 (and references therein) and rigorous treatment of the uncertainties in high frequency nutrient data and its subsequent impact on loads is given by Lloyd et al., 2016. We propose to show 'double banded' plots with one band on the observations to show measurement uncertainty on the discharge and phosphorus load, and one band on the model simulation to show model parametric uncertainty.**

**McMillan, H., Krueger, T., and Freer, J.: Benchmarking observational uncertainties for hydrology: rainfall, river discharge and water quality, Hydrol. Process., 26, 4078-4111, 10.1002/hyp.9384, 2012.**

**Lloyd, C. E. M., Freer, J. E., Johnes, P. J., Coxon, G., and Collins, A. L.: Discharge and nutrient uncertainty: implications for nutrient flux estimation in small streams, Hydrol. Process., 30, 135-152, 10.1002/hyp.10574, 2016.**

Technical corrections
Page 7, line 14: space missing after Avon
**Thanks for noticing these. Space will be added.**
Page 9, line 23+24: The flow fractions for the slow flow component are not really

needed as this are just 100

**This is true, but we prefer to include them so that reader does not have to search back several lines to find the % for the fast components.**

Page 9, line 32: Space missing after "drain flow"

**Space will be added.**

Page 12, line 7: space missing after "have"

**Space will be added before 'have'.**

Page 12, line 31: space missing before "Yang"

**Space will be added.**

---

## Author Comment (AC3) · 25 Aug 2017

Authors' response to Reviewer 2, Anonymous
For clarity, we have included the reviewer's comments in black; our response is in blue

General Comments:
The manuscript "Data-based mechanistic model of catchment phosphorus load improves predictions of storm transfers and annual loads in surface waters" addresses relevant scientific questions and should be published after some major revisions. The highlight of the value of high temporal resolution measurements to capture the system dynamics is very valid, as well as the benefit of the chosen approach in determining the "optimal" temporal resolution for future measurement campaigns. However, especially the "methods" section needs to be expanded by clearly stating and justifying the assumptions made.
**See specific responses below**
Specific comments:
Chapter 2.3:
What is exactly the meaning and the implications of not using a noise model in Eq 2? This should be explained in more detail. Any inference algorithm (in this case probably the RIV(C)BJ), needs to make assumptions about the errors to estimate parameters. Does not using a noise model mean that you assume the errors to be uncorrelated? Or is the error model inferred by the algorithm itself? The assumptions made in the inference process should be clearly stated and checked.
**Noise models were not used for two reasons. (a) full models (input-output (I-O) plus noise) were actually initially evaluated and overall they did not produce better results; therefore, in order to keep a consistent approach for all catchments, noise models were not used in later model identification, (b) transfer function (TF) models with a noise component generally do not improve longer term predictions of processes which are I-O dominated, the noise being modelled as ARMA processes and thus not generating good longer term forecasts. The Reviewer is correct in that the residuals structure was not strong/consistent enough for a noise model to improve the model fit consistently. This will be mentioned in the manuscript.**
Eq 4 leads to significant violation of the mass balance w.r.t. water if $Q(t-1)$ is larger than 1 (this depends strongly on the units of Q) and beta is larger than 0. This should be clearly stated, and then briefly mentioned why this is not a problem in this case (if it is not).
**Sometimes rescaling is applied to Re, to ensure that total Re is equal to total R, and then the TF gain parameters can be interpreted as runoff coefficients. However, if rescaling is not done at that stage, it is balanced by the linear TF parameters (i.e. the rescaling takes place within the transfer function estimation process). In this application, rescaling at the Re stage would not be possible in the 'double layer' TP model, where Re and Q are simulated together, one step at a time.**
Chapter 3:
It would be interesting to validate some models with noise to get the total predictive uncertainty, not just parameter uncertainty. This would be a more meaningful validation and would allow actual statements about the uncertainty related to predictions of TP loads.
**Model residual variance is included in the parametric uncertainty. We propose to show 'double band' plots of model fit, where the data uncertainty will be illustrated as one band on the observations and model parametric uncertainty will be shown, as at present, on the model simulation, thus effectively comparing the distributions at each sample.**
Most plots show observed and modeled quantities in the identification period. It would be interesting to have some more plots in the validation period (ev. including the total uncertainty bands including the noise). Also some zooms to specific time periods showing the strengths and the weaknesses of the models would be interesting.

**We propose to show further plots, including some of validation period, and certainly some zoomed in periods. Thanks for this comment.**

Page 11 L33: The comparison of the quality of fit to process-based models needs a more detailed analysis. How exactly do you compare the quality of fit? With NS coefficients only? Is this justified, are the assumptions underlying NS coefficients fulfilled?

**Direct full comparisons are not currently possible, as the published results for process-based models used different catchments and data sets. Thus only generic model fit comparisons are possible at this time. However, we propose to tone down this comparison with other models, and accordingly we propose a change of title to "Prediction of storm transfers and annual loads with data-based mechanistic models using high-frequency data".**

Technical corrections:

Page 8 L4: should be Table 2

**This is true; thank you for noticing, this will be corrected**

Page 10 L27/28: one of the "fast pathway" should probably mean "slow pathway"?

**No, the text is correct. This paragraph compares the Blackwater TPload model with its corresponding runoff model, so the first 'fast pathway' refers to the proportion of TPload transferred on this pathway (p10, l27) and the second 'fast pathway' refers to the proportion of water transferred (p10, l28)**

Page 11 L22: "in" should be "an"

**Thank you for noticing, this will be corrected**

---

## Author Comment (AC4) · 25 Aug 2017

Authors' response to Reviewer 1, Sebastian Stoll
For clarity, we have included the reviewer's comments in black; our response is in blue

General remarks:
Generally, I find the manuscript to be very interesting, well written and suitable for
HESS (after some revisions).
**Thank you**
While I agree with the authors that DBM models are very
helpful in detecting dominant transfer modes I think that some of the alleged benefits
of the modelling approach are overstated. For example, I doubt that these models can
"help in planning appropriate pollution mitigation measures" as stated in the abstract.
The reason for that is the nature of these models. The only input driving the models is
rainfall (and sometimes discharge) data. Many features which are known to influence
the phosphorus dynamics (like soil type, soil phosphorus concentration, management
practices, tile drainage density, etc.) and which would be the primary entry point for any
mitigation measures are not directly considered. Accordingly, the effect of any changes
in these features (e.g. management practices) cannot be evaluated (not saying that
physically-based models are per se any better with regard to that given the parameter
uncertainty). In my opinion, the presented DBM models are much better suited to
analyze the effects in changes of the precipitation (as rainfall is the main input) under
the condition that these future precipitation conditions are covered in the calibration
period.
**We agree with the reviewer that DBM models in isolation cannot be used directly to evaluate
different mitigation measures, but we believe that the understanding of catchment function, that
for DBM models is** determined from the data alone**, is still helpful in planning appropriate
mitigation measures e.g. targeting of fast/surface pathway in the Newby Beck/Eden catchment.
Indeed, as experiments on various mitigation measures develop and allow for mapping of the
effects of the mitigation measures onto the parameters of the DBM model (e.g. Chappell et al.,
2006 and current projects in Lancaster Environment Centre, Dr N Chappell), the potential for using
DBM to "help in planning appropriate pollution mitigation measures" will increase. Unlike
physically-based models, in which the (generally unknown) structure is fixed beforehand, with
parameters optimised to make it fit the data, the dominant modes of catchment function
determined from DBM models can be interpreted directly. However, we propose to modify the
abstract and text to say "The models led to a better understanding of the dominant transfer
modes, which will be helpful in determining phosphorus transfers following changes in
precipitation patterns in the future."
Chappell, N. A., Tych, W., Chotai, A., Bidin, K., Sinunc, W., and Chiew, T. H.: BARUMODEL:
Combined Data Based Mechanistic models of runoff response in a managed rainforest catchment,
Forest Ecol. Manag., 224, 58-80, 2006.**
In addition, I would love to see some more analysis of the very nice data they collected.
I would assume that the manuscript would greatly benefit if the model results would
be discussed together with the data (for example detailed analyses of the hysteresis
curves).
**We did not include detailed analysis of the high-frequency data as this has already been published
by several authors, e.g. Outram et al., 2014, HESS (including hysteresis analysis); Perks et al., 2015,
Sci. Tot. Environ; Ockenden et al., 2016, Sci. Tot. Environ.**
Specific remarks:
Title: Improvement compared to what, other models? **Yes, this was compared to other models, but
in a general sense only, as direct comparison is not possible unless on the same catchment with
the same dataset. However, we propose to revise the title to "Prediction of storm transfers and**

annual catchment phosphorus loads with data-based mechanistic models using high-frequency data".

P1, L31-32: See comments above

See response above

P2, L7: The authors correctly point out the importance of the measurement uncertainty. However, in the whole manuscript no information is provided regarding the uncertainty of the rainfall, discharge and phosphorus measurements or how this uncertainty is handled in the modelling approach. Especially the stage-discharge relationship (regarding the discharge measurements of flood events) can be subject to considerable uncertainty which would directly translate into uncertainty of the phosphorus loads. One could argue that the measurement uncertainty is indirectly accounted for by the parameter uncertainty. However, given that the uncertainty bands are hardly detectable in the figures and measurements (without error bars) are not covered by it, it seems that either an important process is not captured by the model or that the measurement uncertainty is underestimated.

The figures currently show only the uncertainty resulting from parameter estimation, and with good model fit, that is usually small. We propose to show 'double banded' plots with one band on the observations to show measurement uncertainty on the discharge and phosphorus load, and one band on the model simulation to show model parametric uncertainty.

P2, L24-32: Here, the authors report the disadvantages and shortcomings of large, overparameterized process-based models (e.g. SWAT). I understand the motivation for that and even to a large degree agree with them. However, the authors should not only pick and describe the most extreme (or worst) process-based models. There are also parsimonious process-based models which can deliver reasonable results describing dynamics of phosphorus on hourly time steps (for example Hahn et al., 2013) or spatial herbicides losses (which have very similar transport pathways) (for example Frey et al., 2011) with few parameters.

We have recognised that there is a wide range of models of differing complexity (p2, l17-24) which are applicable in different circumstances. We wanted to contrast the two ends of the scale, which is why we picked on the SWAT model. However, we propose to include some further examples of more parsimonious models as mentioned above: "Less complex process-based models, with fewer parameters, have also been developed for phosphorus transfer and have been applied with reasonable success to specific catchments, e.g. Hahn et al., 2013; Dupas et al., 2016. Both these studies related to small catchments (< 10 km$^2$); it was recognised that the models would only be applicable to locations where the assumptions of the model were satisfied, which is consistent with Beven (2000) and 'uniqueness of place'."

Beven, K. J.: Uniqueness of place and process representations in hydrological modelling, 4, 203-213, 10.5194/hess-4-203-2000, 2000.

P4, L14: What was the motivation to measure TP and not distinguish between or focus on particulate and/or dissolved phosphorus? Particulate (PP) and dissolved phosphorus (DP) can have different pathways and dynamics. While PP often shows a clockwise hysteresis (P peak before Q peak), DP often shows an anti-clockwise hysteresis (Q peak before P peak) (Dupas et al, 2015). By modelling them separately, it would be probably easier to identify a suitable transfer function and the corresponding pathways.

This is a fair point, and ideally we would have looked at both TP and DP/PP. However, we did not have the data for dissolved phosphorus, as none of the bank-side analysis was done on filtered samples. Both TP and Total Reactive Phosphorus (TRP) were measured by the Demonstration Test Catchments teams, who collected the data. It would be interesting to model TRP too (which could be used as an approximation to dissolved reactive P, but we concentrated on TP in this study as the ultimate goal (for the NUTCAT 2050 project, of which this study was part) was to predict TP loads under climate change.

P5, L17: What is R in the equation, rainfall?

**Yes, it is defined on p5, l2.**

P6, L6-10: What is the motivation for setting up these short-term models for the Newby Beck catchment when the long-term model have similar performances and structures?

**The short-term models were linear, i.e. an even more simple structure and even fewer parameters. The purpose was to show that for short periods, where antecedent flows for events were rather similar, a model with just five parameters could be identified. We are evaluating a methodology in this paper, and successful modelling at different time scales can be used as a validation of the approach. This is particularly the case when validating over extreme events – even given the large uncertainty of discharge observations during the selected period (Storm Desmond).**

P6, L15: If I understand it correctly, the output which is used to identify and calibrate the model is also used as an input. I find this contra-intuitive and not really "proper". Why not use a precipitation based antecedent wetness index?

**APIs (antecedent precipitation indices) introduce additional parameterisation, often arbitrary, which is exactly what we are largely avoiding by using DBM methodology. Using a simple nonlinear function (with a single and optimised parameter) of recent discharge measurement as catchment wetness surrogate has been tested on catchments of different size and nature, and published numerous times (e.g. Young and Beven, 1994; Chappell et al., 1999; Young, 2003; McIntyre and Marshall, 2010; Beven, 2012). After all, a recent high flow will be highly correlated with high 'overall' catchment wetness, and using the flow at time t-1, as in Eqn. 4, still allows estimation of Re and Q at time t. A simple antecedent precipitation index was actually tried; it worked with reasonable success for Newby Beck but not for the other catchments, and therefore, as a consistent method was sought for all catchments, the API approach was not pursued in this case. We propose to mention this approach in the manuscript.**

**Young, P. C., and Beven, K. J.: Data-Based Mechanistic Modelling and the Rainfall-Flow Nonlinearity, Environmetrics, 5, 335-363, 1994.**

**Chappell, N. A., McKenna, P., Bidin, K., Douglas, I., and Walsh, R. P. D.: Parsimonious modelling of water and suspended sediment flux from nested catchments affected by selective tropical forestry, 354, 1831-1846, 10.1098/rstb.1999.0525, 1999.**

**Young, P.: Top-down and data-based mechanistic modelling of rainfall-flow dynamics at the catchment scale, Hydrol. Process., 17, 2195-2217, 10.1002/hyp.1328, 2003.**

**McIntyre, N., and Marshall, M.: Identification of rural land management signals in runoff response, 24, 3521-3534, 10.1002/hyp.7774, 2010.**

**Beven, K. J.: Rainfall-runoff modelling : the primer, 2nd edition, John Wiley & Sons, Chichester, 2012.**

P7, L7-10: Some scatter plots would be very helpful to illustrate the Q-P relationships.

**Discharge-TP concentration plots for the three catchments are already given in Supplementary Information Figures S1 – S3. However, we propose to refer to them here as well.**

P7, L16: Table number is missing.

**Thanks for noticing. It should be Table 1.**

P7, L19-20: Because Blackwater has the lowest specific discharge. It would be good to discuss and explain the differences in the specific discharges and P concentrations among the catchments.

**Agreed. We propose to add a column in Table 1 giving the rainfall-runoff ratio for that year, and to change text to say "The lowest mean annual TP concentrations were observed in the Newby Beck catchment, but combined with the highest runoff this resulted in a high total annual TP load. Conversely, although mean annual TP concentration in the Blackwater was also higher than in Newby Beck, when combined with the lowest runoff, this resulted in the lowest total annual TP load. The rainfall-runoff ratio for Newby Beck (0.65) was much higher than for the Blackwater (0.30) or the Wylye (0.32), indicating a larger capacity for storage in the latter two catchments.**

**Despite similarity in the rainfall-runoff ratio, total runoff in the Wylye was higher than the Blackwater because of the higher total rainfall."**

**Differences in the P concentrations are already explained in the paragraph p7, l3-15.**

P7, L28-30: So were model results actually used to fill data gaps for the longterm model? If yes, this should be clearly stated and discussed accordingly.

**The linear model would only have been used to fill data gaps in the short-term data series, if a complete series was required to estimate, for example, TP load over the calibration period, based on observations. This was not actually used in this study. However, the DBM transfer function models can be used in model-based interpolation of the output, given the input signals, just as they can be used in forecasting (e.g. Smith et al, 2014).**

**Smith, P. J., Panziera, L., and Beven, K. J.: Forecasting flash floods using data-based mechanistic models and NORA radar rainfall forecasts, 59, 1403-1417, 10.1080/02626667.2013.842647, 2014.**

P7, L29-30: How can model results help in identifying problems in the extrapolation of the stage-discharge relationships, when the whole model itself is based and calibrated with data of exactly these stage-discharge relationships? In my opinion the model is only reliable for the conditions covered during the calibration period. If more extreme events would be included in the calibration period, the model and the parameters would very likely be different.

**We did not mean to imply that the model could identify problems with stage-discharge relationships, but rather to suggest that it could be useful when there are known problems with the relationship. However, we agree with the comments about model validity outside of the calibration conditions. The calibration (in the sense of stage to discharge) is the weakest point of all the catchment models relying on stage measurements, particularly for extreme events.**

P8, L4: Should be "Table 2"

**This will be changed.**

P8, L4-18: I find the discussion and evaluation of storm Desmond a bit constructed and unnecessary. You don't need a DBM model to realize that discharge and P load was underestimated when there are reports of out-of-bank discharge bypassing the gauging station. The model also doesn't help in the quantification of the missed P and Q. As mentioned before, the model was trained under different conditions and is therefore in my opinion not really valid for very extreme cases not being part of the calibration period (again not saying that physically based models are any better).

**We agree that the results for Storm D are tentative, they are shown here to demonstrate the effectiveness of DBM over a particularly challenging period in data.**

Table 2: According to the time constants and order of the Q- and TP models, there are two pathways contributing to the discharge generation with only the fast pathway contributing to the TP generation. If I understand the concept of the TC correctly, TP reacts before the discharge rises. Is this in agreement with the measured data?

**Shorter time constant in the case of impulse shaped input means that the response grows faster and decays faster, not that it reacts quicker (that would be the time delay, which in this case is the same).**

Table 3: What is the meaning of the term "using Qsim". If model outputs instead of actual measurements were used, this should be clearly stated, justified and discussed (for example why is the performance worse for "using Qsim" that "using Qobs"?) In relation to that, how was TPLoad calculated? Did the authors used the modeled Q to calculate TPLoad or did they use the measured Q? Again, if modeled Q was used, this should be stated, justified and the consequences discussed.

**Thanks for pointing this out – we have not expressed it clearly. The effective rainfall is calculated according to Eqn. 4, using the observed discharge, Qobs, as a proxy for the storage state of the catchment. Model parameters for the linear model (effective rainfall-runoff) are estimated from this. This results in Rt2 using Qobs. However, for a true simulation, Qsim is calculated only from**

**the rainfall and the model parameters, giving Rt2 using Qsim. The effective rainfall – TPload model is a two-stage model; it is assumed that the discharge is unknown, so that the effective rainfall must be calculated one step at a time, as Qsim is generated with the previously identified parameters of the rainfall-discharge model. Hence Rt2 using Qobs is a one-step ahead prediction, whereas Rt2 using Qsim is a true simulation, only using the rainfall input.**
**TPload was calculated according to Eqn. 1, using the observed discharge and the observed concentration.**

Table 3 cont'd: For the Newby and Wylye TPLoad models effective rainfall was used as input, while regular rainfall was used for the discharge model. What is the meaning of that? Does it mean that for TP dynamics antecedent conditions are important, while they are not important for the discharge dynamics? Again, I would advise to discuss these findings as well as the different time constants and their percentages together with the actual measured data.

**Thanks, we realise that our explanation is unclear. All models, apart from the Blackwater rainfall-TP model, are linear models using effective rainfall as input. The effective rainfall is calculated using a non-linear function, according to Eqn. 4. The antecedent conditions are important in both discharge and TP dynamics. The reason the effective rainfall was not used in the Blackwater TPload model is because the simulated discharge, Qsim, is a poor fit (Rt2 using Qsim = 0.37, which is worse than for a rainfall-runoff model with linear rainfall input). We propose to change Table 3 to make it clear that effective rainfall was used in all cases except the Blackwater TPload model**

P9, L26: What does "effective rainfall (from the runoff model)" mean?

**This means effective rainfall calculated one step at a time using Qsim.**

P11, L7: Same point again. What does "effective rainfall simulated by the rainfall-runoff model mean?

**as above**

P12, L1: It's nice to have models with a low parameter uncertainty. However, when the uncertainty bands do not encompass the measurements, it's not really better situation than having a large parameter uncertainty. The model is either missing an important process or measurement uncertainty is not accounted for. A third reason could be a too narrow parameter sampling space in the MC method.

**Parameter sampling used in the MC runs is from a multivariable Gaussian distribution using the estimated parameter values as means and their estimated covariance matrix as covariance. The model fits the data well, so the covariance matrix is small (in L2 sense), and the uncertainty of the model is limited to its parametric uncertainty. What is not accounted for here is the uncertainty of the measurements – see response above. We propose to show figures with double bands – a band on the observations, indicating the measurement uncertainty, and a band on the model simulation, indicating the parameter uncertainty. This will bring the additional value of visual partitioning the uncertainty of model predictions. Thanks for pointing this apparent issue out.**

P12, L28-33: Although, the authors openly discuss the limitations of their modelling approach, there is one point I miss. They argue that understanding the rainfall-Q/TPLoad relationship through DBM models can help to identify dominant modes of the catchment and can therefore be used to target management interventions. I would argue that this is only possible if the identified dominant modes or pathways can be related to specific areas in the catchment. In my opinion it is not enough to know that 70% of the TPLoad was activated via a fast pathway. It is necessary to know which areas in the catchments are connected to the stream via this pathway, how these areas are managed and what their soil P status is. To actually plan and implement intervention strategy, you need to know where (on which fields) and how to intervene. The "how" is strongly dependent on the "where". If you identified some fields with subsurface tile drainage as the contributing areas you would need a different intervention strategy as

for example on a field with a tendency for surface runoff due to soil compaction. Knowing the temporal dynamics is not good enough, you would also need information about the spatial patterns.

**This is true of any model where the observations are from the catchment outlet. It is not possible with any certainty going beyond assertion to apportion the contribution of specific areas without observations characterising these specific areas. We accept that DBM does not provide information about the spatial patterns, but we did not claim that it could be used to "target" management interventions (this makes it sound location specific), merely to be useful in "planning" interventions. We propose to tone down the text in this respect, modifying the abstract to say "The models led to a better understanding of the dominant transfer modes, which will be helpful in determining phosphorus transfers following changes in precipitation patterns in the future."**

Authorship:
I thought long about including this very last comment in the review. However, given the many discussions I had with colleagues about this very issue in the past, I feel somewhat obliged to mention that I find the number of authors contributing to this manuscript too excessive, given the nature of the article (a regular modelling study). I am very much in favor in acknowledging significant contributions (for example with respect to data gathering) with a co-authorship, however this seems not to be the case here. The authors state themselves that while two persons were responsible for the modelling, three persons did project management and the remaining fourteen (!) basically discussed the results and did some editing. I certainly don't want to offend any of the authors and obviously have no insights in the preparation process of the manuscript. Nonetheless, I would encourage each co-author to reflect if in their opinion they really contributed significantly to this manuscript.

**This modelling study contributed to a large consortium project (NUTCAT 2050), the ultimate aim of which was to make predictions of phosphorus transfer into the future. As part of that project, this modelling was developed and discussed with the project team, alongside other modelling approaches. All members of the NUTCAT 2050 team have been involved in the evolvement of this modelling study and have contributed to the manuscript. Other co-authors were involved with the Demonstration Test Catchment (DTC) Project, which collected the data. To clarify, we propose to add to the author contributions "MCO, KJB, ALC, RE, PDF, KJF, KMH, MJH, RK, CJAM, MLV, CW, PJW, JGZ and PMH contributed to NUTCAT 2050; ALC, KMH, SB, RJC, JEF and PMH are part of the DTC project."**

References
Dupas, R., Gascuel-Odoux, C., Gilliet, N., Grimaldi, C., Gruau, G., 2015. Distinct export dynamics for dissolved and particulate phosphorus reveal independent transport mechanisms in an arable headwater catchment. Hydrol. Process. 29 (14), 3162e3178. http://dx.doi.org/10.1002/hyp.10432.
Frey,M.P.; Stamm,C.; Schneider,M.K.; Reichert,P. (2011) Using discharge data to reduce structural deficits in a hydrological model with a Bayesian inference approach and the implications for the prediction of critical source areas, Water Resources Research, 47(12), W12529 (18 pp).
Hahn,C.; Prasuhn,V.; Stamm,C.; Lazzarotto,P.; Evangelou,M.W.H.; Schulin,R. (2013) Prediction of dissolved reactive phosphorus losses from small agricultural catchments: Calibration and validation of a parsimonious model, Hydrology and Earth System Sciences, 17(10), 3679-3693.

---

## Author Response (AR1)

**Response to Editor's comments HESS 2017-314**
**For clarity we have included the editor's comments in black; our response is in blue**

**Dear Dr Stamm,**
**Thank you for the Editor's comments. We have responded in detail below and revised the manuscript to reflect both your comments and those of the referees.**

Data-based mechanistic model of catchment phosphorus load improves predictions of storm
transfers and annual loads in surface waters
Editor comments to the response to the reviews
8.9.2017
Dear Dr. M. C. Ockenden
Thank you for the responses to the comments provided by the reviewers.
In most cases, I consider the suggested modifications of the manuscript and/or the replies as
satisfactory. However, regarding the issue of model uncertainty the revision needs to go beyond
what you suggested. Reviewer 1 and 2 raised some important questions that need more in-depth
responses and analysis in the manuscript.

In the Introduction you mention explicitly that even though the model structure is largely data
driven, structural model errors will remain (p. 5, L. 10). You also discuss limitations of model
structures in section 3.5 (p. 12, L: 6 – 15). Despite acknowledging model structure as a source of
uncertainty, you almost completely ignore this aspect when responding to comments by the
reviewers addressing this aspect. Reviewer 1 for example comments "…*However, when the
uncertainty bands do not encompass the measurements, it's not really better situation than having a
large parameter uncertainty. The model is either missing an important process or measurement
uncertainty is not accounted for.*".
**We have been open about the structural errors in the DBM model, mentioning them in the introduction, and in section
3.5. Indeed, it is part of the DBM modelling technique that one is prepared to accept simplification of a complex system
(i.e. a structure which captures only the dominant modes and not every single process) in order to reduce parameter
uncertainty. We have chosen not to show the residual error in plots, as the main focus of this paper is to explore the
methodology and the novelty of modelling TP load with high-frequency data using the DBM technique. Of course
uncertainty is important in that, but the beauty of this type of modelling is that very little of the uncertainty is due to
the parameters (partly because there are so few of them). One could attribute some of the mismatch in model and
observations to structural uncertainty, but at least we know that the model captures the dominant processes that are in
the data. We disagree with the comment that "when the uncertainty bands do not encompass the measurements, it's
not really better situation than having a large parameter uncertainty." Agreed, if total uncertainty or model fit statistics
are all you are interested in, then there is little difference, but with this technique (with its acceptance of structural
simplification) it means that the dominant modes are identified from the data alone. These dominant modes are given a
physical interpretation and therefore aid in catchment understanding. Indeed, most calibration exercises (including
simply fitting a regression to data) are carried out conditional on an assumption that the model is correct and that any
structural error will be implicit in the model residuals. Structural error is an epistemic error and cannot be allowed for
directly.**

**We have added a new section on uncertainty (section 2.4). We have added measurement data uncertainty to the plots
so that the relative magnitudes of model parameter uncertainty and measurement data uncertainty can be seen. We
have chosen not to show total model predictive uncertainty on plots as this does not add to the message of this paper,**

**but we have added a note in the caption of each figure pointing out that the residual uncertainty (not shown) adds to the total predictive uncertainty.**

You reply that "…The model fits the data well, so the covariance matrix is small (in L2 sense), and the uncertainty of the model is limited to its parametric uncertainty."

**We did not express ourselves clearly here. We meant to imply that an identified model (not including any error associated with the structural simplification, which is accepted as part of this technique) has low parametric uncertainty because the linear-dynamic part of the process that the model describes is well-defined (see also comments below). We are well aware of the error associated with model structure, but accept it as part of this technique. For consistency, we have removed the one statement which mentioned model predictive uncertainty (previously p5, l30-31) but have noted in each figure caption that the total predictive uncertainty (which includes the residual uncertainty) is larger than the parametric uncertainty and would enclose the observations most of the time.**

First, you do not comment at all at the correct observation by the reviewer that for a substantial fraction of time the uncertainty bands do not enclose the observations (see Fig. 4 – 6). Second, you explicitly claim that parameter uncertainty dominates the model predictive uncertainty ("…The model fits the data well, so the covariance matrix is small (in L2 sense), and the uncertainty of the model is limited to its parametric uncertainty."). Implicitly you also suggest that by showing the measurement uncertainty the deviations between observations and simulations can be explained.

**Again, we did not express ourselves clearly. We did not mean to imply that the measurement uncertainty alone would explain the deviations (see above) or that the parametric uncertainty dominates the model predictive uncertainty. The input-output (I-O) model that the DBM method produces has low parametric uncertainty because the linear-dynamic part of the process that the model describes is well-defined. However, the overall observations are described by the output of the linear-dynamics driven by the rainfall observations PLUS all the other catchment process with their associated uncertainties, including measurement uncertainty of both the input (catchment rainfall estimates are affected by non-homogeneity of the rainfall field and rainfall regime) and output. These other processes are not identifiable from the available data, and therefore we cannot reliably quantify them using DBM or any other method. We accept that, by having small parameter uncertainty, this shifts a larger part of the total uncertainty to the residual uncertainty (including measurement uncertainty, unmeasured inputs and some element of structural uncertainty), but an accurate partition of uncertainty was never the aim of this work, which was to build a unified approach to DBM modelling of TP load in the three catchments.**

**We have added measurement data uncertainty to the plots so that the relative magnitudes can be seen. We have added text commenting on the mismatch between models and observations, and have expanded sections of the figures to illustrate particular cases. We have also noted in each figure caption that the total predictive uncertainty (not shown, but which includes the residual uncertainty) is larger than the parametric uncertainty and would enclose the observations most of the time.**

However, you do not provide any arguments why model structure was not a relevant source of uncertainty.

**We have added a new section on uncertainty (section 2.4), including why model structure is a source of error (see also our response above). We did not mean to imply that it was not. We feel that focussing on the uncertainty (which may be more important for process-based models where one is unsure exactly what the uncertainty is due to) detracts from the central focus of this paper, which is exploring a different methodology.**

In addition, one has to consider that you actually skipped the error term in Eq. 2. What are the implications for uncertainty quantification?

**We have modified the text to indicate that we did not include a specified noise model. However, if a noise model is not specified, the algorithm includes a default noise model which assumes normally distributed, uncorrelated errors. We**

**accept that this is unlikely to be the case, especially with high frequency data where under/over-prediction at time t is likely to result in under/over-prediction at time t+1, but it is often a reasonable approximation. Although residual analysis (included in the Supplementary Information) indicates both autocorrelation and heteroscedasticity in the residuals, we found that the use of an ARMA structured noise model did not improve results overall for all three catchments. In order to keep a consistent method for all three catchments, a structured noise model was not included. In trying to make improvements for a specific catchment, this is one area that could be investigated further.**

In addition to this lack of actual evidence for your statements, it also contradicts other statement in the text and the data you present. On p. 5, L. 30 – 31 you write: "Prediction bounds for the model can be calculated by adding the residual uncertainty and the parameter uncertainty." Comparing the deviations between simulations and observations in Fig. 4 for example with the magnitude of the indicated parameter uncertainty makes it hard to reconcile them with your statements above. This holds true even when considering the aspect of measurement uncertainty. You briefly touch upon that issue in the text and suggest to add this information to the (figures in) the revised version. While this a very valuable suggestion, a closer visual inspection of the data casts doubts whether measurement uncertainty can fully explain the mismatch between observations and model simulations.

**Measurement uncertainty alone will not explain all the mismatch between observations and the model, and we never meant to imply this. We have added text commenting on the mismatch between models and observations, and have expanded sections of the figures to illustrate particular cases. We have removed the statement about "prediction bounds for the model…" as we have not included the residual uncertainty in the figures, in order to be able to see the relative magnitudes of the parameter uncertainty and measurement data uncertainty. We have also noted in each figure caption that the total predictive uncertainty (which includes the residual uncertainty) is larger than the parametric uncertainty and would enclose the observations most of the time.**

In summary, there are two aspects where you do not provide a satisfactory answer to important questions of the reviewers regarding model uncertainty. First, throughout the text you deal with the different sources of uncertainty in an inconsistent manner. In some paragraphs, model structure is considered, in others not, at some places residual errors are explicitly mentioned, later on they are completely ignored. Second, there are obvious discrepancies between the model predictions and simulations that ask for i) a proper presentation of the relevant data (e.g. the residuals) and ii) a coherent discussion of possible sources of uncertainty. Please note, the issue is not that the model results were not of sufficient quality. It is only about the presentation and the discussion of the actual uncertainty.

To address these issues, I ask you to provide the following data and information:
- Refer to the different sources of uncertainty in the Introduction (and Method section) and explain how you have quantified and /or accounted for them in the context of this paper. Please also refer to input uncertainty.

**Different sources of uncertainty are mentioned in the Intro (para 3) and in the Methods section (new section 2.4 on p8), including how these were quantified**

- Description how the measurement uncertainties were quantified. This has to include an explanation how you dealt with the mix of systematic and random measurement errors and how you accounted for the temporal auto-correlation of discharge errors.

**A new section 2.4.3 (p9) details how measurement uncertainties were quantified**

- Provide the uncertainty bands for the observations (as suggested in your response).

**These have been added to figures.**

- Provide an analysis of the residuals (for discharge and TP) by showing time series of the residuals and the residuals as a function of the observed values. This can go to the SI but the

reviewers and readers need this data to judge the model quality.

**Time series of residuals and residuals against observed values have been added to SI (Newby discharge model SI Fig. S9; Newby TP load model SI Fig. S10; Wylye discharge model SI Fig. S11; Wylye TP load model SI Fig. S12; Blackwater discharge model SI Fig. S13; Blackwater TP load model SI Fig. S14).**

- Explain how measurement uncertainty can explain systematic deviations between simulations and observations given the fact that you fit the model to these observations.

**We did not mean to imply that the measurement uncertainty alone would explain the deviations (see above). We have added measurement data uncertainty to the plots so that the relative magnitudes can be seen. We have not shown residual uncertainty on plots, but have noted in each figure caption that the total predictive uncertainty (which includes the residual uncertainty) is larger than the parametric uncertainty and would enclose the observations most of the time. We have added text commenting on the mismatch between models and observations, and have expanded sections of the figures to illustrate particular cases.**

- When making statement about the origin of uncertainty, please refer to data such that readers can follow your arguments by referring to actual data (e.g. the residual analysis).

**The residual analysis is included in the SI, and reference is made to these at several points in the text.**

Based on the visual inspection of e.g., Fig. 4 or 6, I have the impression that peaks (of a certain magnitude) are systematically underestimated (I am happy to see if your data proofs me wrong). Again, should this be the case, I don't see any problem in that for the manuscript, but think that this was an important aspect to know. First, from a theoretical point of view one would ask why a DBM could not capture that? Would even longer time series be needed? Such aspects would nicely fit into Chap. 3.5. But also from in practical terms it might useful because it could pinpoint hydrological conditions during which pronounced TP loads/concentrations occur (even if they are not fully reproduced by the model).

**It is true that larger peaks have generally been underestimated by the models here. This may be due to the longer time series used, which makes the model applicable over a wider range of conditions, but necessitates the use of the non-linear rainfall input to reflect the storage state of the catchment. It could be that the calculation of effective rainfall (Eq. 6) is not particularly appropriate under the wettest hydrological conditions, when the catchment may be temporarily saturated and acting in a more linear fashion than Eq. 6 allows. Alternatively, it could be due to error in the catchment rainfall estimate (based on three rain gauges) used as input; during particularly large storms (and at other times too), the true catchment rainfall may be affected by the non-homogeneity of the rainfall field, perhaps by localised rainfall on high ground. These uncertainties are not easily quantifiable by any method and apply to any modelling technique.**

There were two additional comment that were only partially answered. Reviewer 2 asked "*What is exactly the meaning and the implications of not using a noise model in Eq 2? This should be explained in more detail. Any inference algorithm (in this case probably the RIV(C)BJ), needs to make assumptions about the errors to estimate parameters. Does not using a noise model mean that you assume the errors to be uncorrelated? Or is the error model inferred by the algorithm itself? The assumptions made in the inference process should be clearly stated and checked.*"

I think your response does not really provide the answer to what the reviewer wanted to know. As I interpret the comment it was not just about why you skipped the error term in Eq. 2. The point is that in order to estimate your model parameters the simulated and the observed values are compared. If you use all the hourly values to minimise your objective function you implicitly assume that all these data points are independent and uncorrelated. In reality however, the observed and simulated values are highly auto-correlated (for some typical time scale). If you model prediction is overpredicting discharge at time x, it is highly probable that the same holds true for the next time step x+1 (as a consequence of your high temporal resolution). For these reasons, people have tried to base their parameter estimates on innovation for example (e.g., Yang, Reichert et al. 2007). I think the reviewer wanted a clarification of that aspect.

**We have clarified that, when no noise model is specified, a default white noise model is used. We have also added a comment in the limitations section 3.5 to suggest that an autoregressive error model may improve model fit for specific catchment applications.**

*Reviewer 2 also stated: "Eq 4 leads to significant violation of the mass balance w.r.t. water if Q(t-1) is larger than 1 (this depends strongly on the units of Q) and beta is larger than 0. This should be clearly stated, and then briefly mentioned why this is not a problem in this case (if it is not)."*

I have to admit that I could simply not follow your argument and see how the water balance problem is avoided. Can you rephrase?

**A transfer function model is not subject to a direct mass balance constraint, and may even seek to relate input and output in different units. The (indirect) mass balance comes through the identified parameters. We have reworded the text on p7 to rephrase.**

In summary, I ask you to revise the manuscript according your responses that you have provided to each review and to additionally address the issues that I have listed above.

Sincerely

Christian Stamm

Editor HESS

References:

Yang, J., P. Reichert, K. C. Abbaspour and H. Yang (2007). "Hydrological modelling of the Chaohe Basin in China: Statistical model formulation and Bayesian inference." Journal of Hydrology **340**(3-4): 167-182.

**We have made the requested changes and hope that you will now find the manuscript acceptable for publication.**

**Yours sincerely,**
**M. C. Ockenden**
**16.10.17**

[revised manuscript text omitted]

*where effective rainfall is used as input to the linear DBM discharge model, this is estimated at the same time as the model parameters, using rainfall R as input

**where effective rainfall is used as input to the linear DBM TPload model, this is first calculated using the previously estimated parameters for the discharge model

Figure S1

Hourly streamflow (Q) against total phosphorus (TP) concentration for the Newby Beck catchment, with the rising limb of storm hydrographs in blue and the falling limb of hydrographs in red.

[Figure]

Figure S2

Hourly streamflow (Q) against total phosphorus (TP) concentration for the Blackwater catchment, with the rising limb of storm hydrographs in blue and the falling limb of hydrographs in red.

[Figure]

Figure S3

Hourly streamflow (Q) against total phosphorus (TP) concentration for the Wylye catchment, with the rising limb of storm hydrographs in blue and the falling limb of hydrographs in red.

[Figure]

Hourly streamflow (Q) against total phosphorus (TP) load for the Newby Beck catchment, with the rising limb of storm hydrographs in blue and the falling limb of hydrographs in red.

[Figure]

Hourly streamflow (Q) against total phosphorus (TP) load for the Blackwater catchment, with the rising limb of storm hydrographs in blue and the falling limb of hydrographs in red.

[Figure]

Figure S6

Hourly streamflow (Q) against total phosphorus (TP) load for the Wylye catchment, with the
rising limb of storm hydrographs in blue and the falling limb of hydrographs in red.

[Figure]

[Figure]

[Figure]

Time series of residuals and histogram of residuals for Figure 4

[Figure]

[Figure]

Discharge model, Newby Beck: Time series of residuals (top); residuals against discharge per unit area (bottom)

[Figure]

[Figure]

TP load model, Newby Beck: Time series of residuals (top); residuals against TP load (bottom)

[Figure]

[Figure]

[Figure]

[Figure]

TP load model, Wylye: Time series of residuals (top) residuals against TP load (bottom)

[Figure]

[Figure]

Discharge model, Blackwater: Time series of residuals (top); residuals against discharge per unit area (bottom)

[Figure]

[Figure]

TP load model, Blackwater: Time series of residuals (top); residuals against TP load (bottom)

[Figure]

[Figure]

Discharge model (a) and TP load model (b) for validation period, Newby Beck

[Figure]

[Figure]

Discharge model (a) and TP load model (b) for validation period, Wylye

[Figure]

[Figure]

Discharge model for calibration period, Blackwater, where effective rainfall has been generated using Qobs (observations) (a) and using Qsim (simulation) (b), showing the poor fit which made Qsim unusable in the TPload model.

[Figure]

[Figure]

Discharge model (a) for validation period, Blackwater, showing poor fit which made effective rainfall unsuitable for use in the TP model; and TP model (b) for validation period using linear rainfall input

[Figure]

Second-order discrete-time model,
a = 1.0000 -1.9324  0.9325 ; b = 0.0000  0.0000  0.0000  0.0000  0.0000  0.0000  0.0526 -0.052
Rt2 = 0.32

[Figure]

Second-order model, linear rainfall :
a = 1.0000  0.0826  0.0002 ; b = 0.0335  0.0002
Rt2 = 0.31

Supplementary references

Franklin, G. F., Powell, J. D., and Emami-Naeini, A.: Feedback Control of Dynamic Systems, 4th Edition, Prentice-Hall, 2002.

UKCP09: Gridded observation data sets: http://www.metoffice.gov.uk/climatechange/science/monitoring/ukcp09/ access: 18 August 2015, 2009.

Ockenden, M. C., Hollaway, M. J., Beven, K., Collins, A. L., Evans, R., Falloon, P., Forber, K. J., Hiscock, K. M., Kahana, R., Macleod, C. J. A., Tych, W., Villamizar, M. L., Wearing, C., Withers, P. J. A., Zhou, J. G., Barker, P. A., Burke, S., Freer, J. E., Johnes, P., Snell, M. A., Surridge, B. W. J., and Haygarth, P. M.: Major agricultural changes required to mitigate phosphorus losses under climate change, Nat Commun, 10.1038/s41467-017-00232-0, 2017.

Robson, A., and Reed, D.: Flood Estimation Handbook -  FEH CD-ROM 3, Institute of Hydrology, Wallingford, 1999.

Soil Survey of England and Wales: Legend for the 1:250,000 Soil Map of England and Wales, Soil Survey of England and Wales, Rothamsted Experimental Station, Harpenden, 1983.

Young, P. C.: Recursive Estimation and Time-Series Analysis, Springer-Verlag, Berlin, 1984.

---

## Author Response (AR2)

**Dear Dr Stamm,**

5  **Thank you for the Editor's comments.  We have responded below and revised the manuscript accordingly.**
**Yours sincerely,**
**Mary Ockenden**
**7.11.17**

10  Editor Decision: Publish subject to minor revisions (review by editor) (26 Oct 2017) by Christian Stamm
Comments to the Author:
HESS 2017-314

Data-based mechanistic model of catchment phosphorus load improves predictions of storm transfers and annual
15  loads in surface waters

Editor comments, 26.10.2017

Dear Dr. M. C. Ockenden

Thank you for the responses to the previous comments and the modification of the manuscript and the
Supplementary Material. This has improved the manuscript substantially, clarified many issues and provides very
useful information.
**Thank you**

There is one single aspect where I see a need for further improvement before the manuscript is ready for
publication. It relates to underestimation of the large TP loads (see Fig. S10, S12, S14) and the explanation you
provide in your response (p. 4). You mention the length of the time series and the possibility for errors in rainfall
estimates as potential causes. However, these arguments are not fully convincing because the discharge is much
30  less affected by patterns of the residuals (compare e.g. Fig. S9 and S10). This suggests that there is an issue
how the P mobilisation is simulated for such events.
**We have added a paragraph to acknowledge the underestimation of large TP loads and suggest ways in which this could
be improved (see below).**

35  This observation raises two questions: First, how does it come that a DBM model (provided the available data
set) cannot capture this systematic pattern? Would a more complex model capture such a pattern but at the costs
of too many parameters? Providing a (tentative) answer to this question can help readers to better understand
the potential and limitations of the DBM approach. Second, what could be the physical or chemical process that
was not properly captured or what was the simplification that goes with the model? Your thoughts and insights on
40  that aspect may be highly relevant for further studies to include processes that may have been overlooked in the
past or to represent them in a better way. I think a few comments on those two aspects would fit well into the first
paragraph of 3.5 and further add value to your manuscript.
**We suggest that the non-linearity in the TP load model may be rainfall, discharge or load dependent, and that Equation
6 may not be capturing this adequately.  Higher order models, when tried initially, did not make a significant
45  improvement and added unnecessary parameters.  We suggest that a flushing effect could be accounted for with models

[revised manuscript text omitted]